# Theta-phase dependent neuronal coding during sequence learning in human single neurons

Leila Reddy [1,2,3,9 ✉], Matthew W. Self [4,9], Benedikt Zoefel[1,2], Marlène Poncet [1,2], Jessy K. Possel [4], Judith C. Peters [4,5], Johannes C. Baayen[6], Sander Idema[6], Rufin VanRullen [1,2,3] & Pieter R. Roelfsema [4,7,8]

The ability to maintain a sequence of items in memory is a fundamental cognitive function. In the rodent hippocampus, the representation of sequentially organized spatial locations is reflected by the phase of action potentials relative to the theta oscillation (phase precession). We investigated whether the timing of neuronal activity relative to the theta brain oscillation also reflects sequence order in the medial temporal lobe of humans. We used a task in which human participants learned a fixed sequence of pictures and recorded single neuron and local field potential activity with implanted electrodes. We report that spikes for three consecutive items in the sequence (the preferred stimulus for each cell, as well as the stimuli immediately preceding and following it) were phase-locked at distinct phases of the theta oscillation. Consistent with phase precession, spikes were fired at progressively earlier phases as the sequence advanced. These findings generalize previous findings in the rodent hippocampus to the human temporal lobe and suggest that encoding stimulus information at distinct oscillatory phases may play a role in maintaining sequential order in memory.

[1] Université de Toulouse, Centre de Recherche Cerveau et Cognition, Université Paul Sabatier, Toulouse, France. [2] CNRS, UMR 5549, Faculté de Médecine de Purpan, Toulouse, France. [3] Artificial and Natural Intelligence Toulouse Institute (ANITI), Toulouse, France. [4] Department of Vision and Cognition, Netherlands Institute for Neuroscience (KNAW), Amsterdam, The Netherlands. [5] Department of Cognitive Neuroscience, Faculty of Psychology and Neuroscience, Maastricht University, Maastricht, The Netherlands. [6] Amsterdam University Medical Centers, location VUmc, Departments of Neurophysiology and Neurosurgery, Amsterdam, The Netherlands. [7] Department of Integrative Neurophysiology, Centre for Neurogenomics and Cognitive Research, Vrije Universiteit, Amsterdam, The Netherlands. [8] Psychiatry Department, Academic Medical Center, Amsterdam, The Netherlands. [9]These authors contributed equally: Leila Reddy, Matthew W. Self. ✉email: leila.reddy@cnrs.fr

Learning to memorize and maintain a sequence of items in memory is fundamental for successful behavior. At the cellular level, this process is linked to sustained firing of individual neurons during the delay period after a stimulus[1–5], as well as by anticipatory activity before the onset of an expected stimulus[6]. Short-term memory processes have also been linked to brain rhythms[7,8], and there is converging evidence that neuronal firing activity during short-term memory can be timed (or phase-locked) to theta oscillations. In humans, the strength of theta phase locking is predictive of human memory strength[9] and navigational goals[10]. In rodents, spiking activity of place cells[11] is locked to specific phases of the theta rhythm during spatial navigation. As a rat runs through a sequence of spatial positions, place cells that represent each of these locations fire at distinct phases of the underlying theta rhythm, and their spikes occur at increasingly early phases as the rat traverses the neuron's place field. This process has been called phase-precession and has been proposed to play a role in the learning of a sequence of spatial positions in the rodent brain[12]. It is unknown, however, if phase precession of action potentials occurs in humans[13].

Here we asked whether a theta phase-dependent coding scheme occurs when human subjects learn a sequence. Because humans are more visual creatures than rodents, we assumed that learning a sequence of visual objects could be analogous to learning a sequence of spatial positions in rodents. We hypothesized that theta phase could be observed during stimulus encoding, and that the phase at which a given cell fires would vary with stimulus identity and order. In other words, while participants are involved in learning a stimulus sequence, each item in the sequence might be represented by neuronal activity that is locked to a different theta phase. Will medial temporal lobe (MTL) neurons in human participants that navigate a "conceptual" space defined by a sequence of visual stimuli exhibit the same form of phase precession that has been observed in rodents during spatial navigation? Do MTL neurons also fire at increasingly early phases when the participant approaches the concept that best activates the cells?

## Results

To create an analogy with the navigation of spatial positions in rodents, we designed a conceptual space that consisted of a sequence of images (Fig. 1A, B). In this conceptual space, images were displayed on the rim of a rotating "wheel" that moved in the clockwise direction. The wheel moved forward smoothly during the inter-stimulus interval (ISI; 0.5 s), during which period a gray placeholder covered all the images. At the end of the ISI period, the wheel stopped for 1.5 s, and the placeholder at the topmost position of the wheel was replaced by the next image in the sequence. We incentivized the participants to learn the sequences by including probe trials (20% of trials). On probe trials we did not present the next stimulus, but showed two choice stimuli instead, and the observers had to indicate which stimulus was next in the sequence. In our previous work with the same paradigm[6], we showed that human MTL neurons that initially responded to a particular ("preferred") image on the wheel started firing in anticipation of this preferred image as a result of sequence learning, during the immediately preceding stimulus and the intervening ISI (Fig. 1D). This finding is reminiscent of rodent place cells that show anticipatory activity in sequentially ordered spatial environments[14]. In the current study, we ask whether theta-phase precession observed in rodent place cells also occurs when humans navigate this conceptual space. In other words, are different stimuli in the sequence assigned a particular theta phase for firing, and is the order of theta phases similar to that observed in rodent phase precession?

Nine human participants learned the order of a fixed number of stimuli (5−7) presented in a pre-defined sequence, while we recorded spiking and LFP activity from 551 neurons in the hippocampus and temporal cortex. Participants rapidly learned the sequence order (>90% performance on probe trials within 6 sequence presentations[6]), and consequently all trials except the probe trials were included in the analyses. For each neuron we identified a preferred stimulus, and the stimuli before and after the preferred stimulus in the sequence were labeled as the preceding and following stimuli, respectively (Fig. 1B). We then compared the theta-phase of firing for the preceding, preferred and following stimuli, and determined whether neuronal responses elicited by these stimuli are encoded at distinct theta phases, reflecting the order in which the sequence unfolds.

Typically, in the human hippocampus, a relatively small subset of "strongly-tuned" neurons (<20% of recorded neurons) shows strong and sparse visual selectivity for particular stimuli[15], and the remaining neurons are frequently omitted from further analysis. However, recent work shows that such weakly-tuned neurons can participate in a reliable ensemble neural code[16,17]. In the current study, we also included these neurons. For each cell, we identified the preferred stimulus as the stimulus in the sequence that elicited the largest number of spikes (i.e., the "hot spot" in the sequence, Fig. 1C). The reliability of each neuron's stimulus preference was verified with a cross-validation analysis (see Methods); only cells that showed a consistent stimulus preference were included for further analyses ($N = 452$). Of these, 102 (~22.5%) were "strongly-tuned" (i.e., their response covaried significantly with stimulus identity as determined by an ANOVA, $p < 0.05$), and we refer to the rest as "weakly-tuned". This approach for neuron selection allowed us to retain a large number of cells in our population (~80%). Supplementary Figs. 3–8 shows examples of individual cells, and Supplementary Fig. 13 shows that the main results were replicated across "strongly-tuned" and "weakly-tuned" cells.

In accordance with previous studies[9,10,18,19], we observed prominent oscillatory activity in the LFP power and the spike-triggered power (STP) across all recording electrodes in the theta (4−8 Hz) and beta (10−18 Hz) bands (Fig. 2 A, B, Supplementary Fig. 1). Neuronal spiking activity was significantly phase-locked (Rayleigh test, $p < 0.0005$) in both frequency bands (Fig. 2C), with a preferred firing phase around the trough of the oscillation (pi radians).

We next determined whether the neurons encoded the preceding, preferred, and following stimuli at different phases of the theta and beta cycles. The neurons fired at distinct phases for these stimuli in the theta band (Fig. 3A, B; Supplementary Fig. 2, Supplementary Figs. 3−8 for individual cell examples). In contrast, the phase of spikes relative to beta oscillations (Fig. 3C, D) was relatively constant during the sequence despite a strong beta band oscillatory component. The pair-wise difference in phase preferences in the theta band for the different stimulus types was significant across cells ($F(1,903) = 37.2$; $p < 0.0001$ for preferred vs. preceding; $F(1,903) = 16.4$; $p < 0.0001$ for preferred vs. following; and $F(1,903) = 74.8$; $p < 0.0001$ for preceding vs. following; Watson−William test significant with Bonferroni-correction for three comparisons).

Importantly, similar to the phenomenon of phase precession in the rodent hippocampus, phases advanced to earlier values as the sequence progressed (mean angle ± standard deviation across cells = 233 ± 35° (preceding), 181 ± 16° (preferred), and 117 ± 9° (following). This corresponds to an average phase difference between consecutive stimuli of −58 degrees/stimulus. This phase lag, measured at the individual cell level rather than the population level, remains significantly negative (−19.3 degrees/stimulus, circular median test against 0, $p = 0.01$; Supplementary

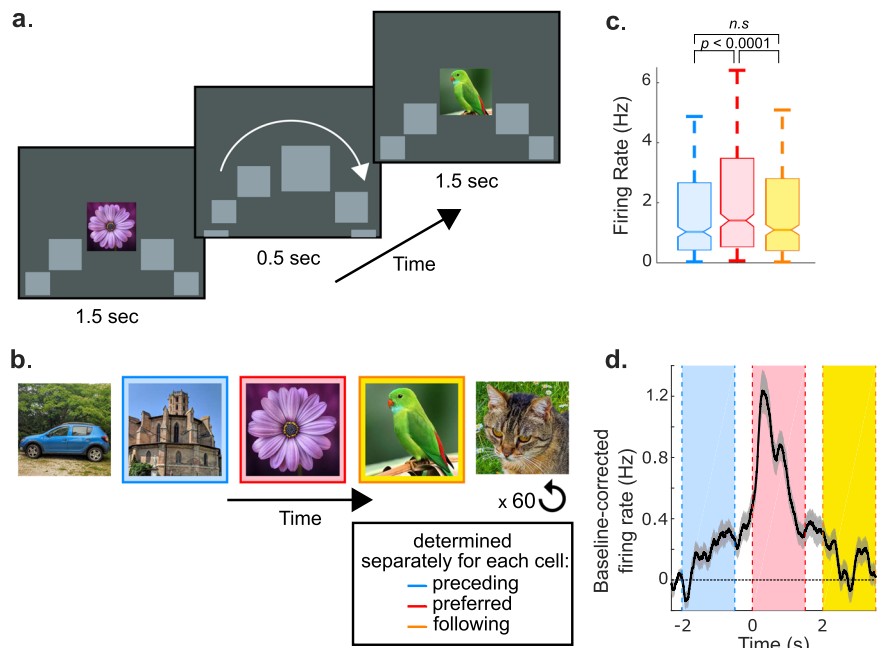

**Fig. 1 Experimental Design. A** In the sequence learning experiment, the participants saw a sequence of 5-7 images in a fixed order. Each image was presented for 1.5 s followed by an ISI of 0.5 s. During the image presentation period (1.5 s), an image was presented at the center of the screen, flanked by gray placeholders. During the ISI period, a placeholder replaced the central image, and all the placeholders moved in the clockwise direction. At the end of the ISI period, the central placeholder was replaced by the next image in the sequence. **B** An example of a 5-image sequence. The sequence was repeated 60 times in each experimental session. **C** For each neuron ($N = 452$) we determined the image that elicited the most spikes. This image was designated as the preferred image for that neuron. Stimulus preference was verified with a cross-validation analysis (see Methods). The images before and after the preferred image in the sequence were designated as the preceding and following images, respectively. The preferred image had the highest firing rate; the responses to the preceding and following images were not significantly different from each other ($p = 0.8$ paired t-test, two-sided). In the box and whisker plots, the central mark indicates the median firing rate across cells, the notch reflects the 95% confidence interval for the median, and the box reaches from the first to the third quartile (interquartile range). **D** Anticipatory responses observed during sequence learning. The mean firing response across all cells ($N = 452$) is plotted as a function of time. Time = 0 is the onset of the preferred stimulus for each cell, time = −2 s is the onset of the preceding stimulus and time = 2 s is the onset of the following stimulus. The blue, pink and yellow shaded areas correspond to the stimulus periods of the preceding, preferred and following stimuli. The white regions between stimuli are inter-stimulus intervals. Mean firing activity was corrected by subtracting baseline activity before the preceding stimulus (−2.5 s to −2 s). Note the anticipatory firing prior to the onset of the preferred stimuli, as a gradual increase of activity during the preceding stimulus[6].

Fig. 2). Phase precession implies that the cycle of action potentials is slightly shorter than the theta band cycle in the LFP. It predicts a slightly higher frequency in spiking compared to the LFP theta oscillation[20] and this is indeed what we observed (Supplementary Fig. 9). To examine how the preferred phase evolved at a finer time scale as the sequence progressed from one stimulus to the next, we plotted the mean phase across cells as a function of time (Fig. 4, Supplementary Figs. 3−8 for individual examples). We observed a change in the phase during successive stimuli. The phase difference between the two extreme stimuli (preceding vs. following) was significant in the theta band (60°/stimulus, Watson-William test, F(1,903) = 48.3, $p < 0.001$), and just significant in the beta band (12°/stimulus, Watson−William test, F(1,903) = 5.5, $p < 0.05$).

The phase precession observed for the preceding, preferred and following stimuli was not observed for the remaining stimuli in the sequence, indicating that the phase selectivity did not extend to stimuli that were far away from their preferred stimuli in the sequence (Supplementary Fig. 2A). This decrease in phase locking is not unexpected if the preferred stimulus reappears every 5−7 stimuli, because items that follow the preferred stimulus will, at some point, be perceived to precede its next presentation, causing a breakdown of phase coding.

We carried out a number of control analyses to rule out the possibility that the phase differences were caused by (1) differences in firing rates, (2) differences in LFP power, (3) a phase

reset around stimulus onset, or (4) differences in theta power. First, we considered the possibility that a difference in the number of spikes fired for each stimulus was responsible for the phase differences (Supplementary Fig. 2 and Supplementary Fig. 10). However, the largest phase difference was observed between the preceding and following stimuli ($p < 0.0001$), even though the firing rates were not significantly different from each other (Supplementary Fig. 2A; $p = 0.8$). Furthermore, the preferred stimuli elicited a significantly higher firing rate than both preceding and following stimuli, yet their spike phase was intermediate. It is thus unlikely that a difference in firing rates could explain the phase precession effect. We also matched the spike count of individual trials between preceding and following stimuli, only including the subset of trials for the preceding and following stimuli in which the difference was <=1 spike (Supplementary Fig. 10), and observed that phase differences were maintained (F(1,842) = 36.2; $p < 0.0001$; Watson−William test). We can therefore be confident that the difference in phase is a robust finding that is not caused by differences in the firing rate elicited by successive stimuli. Phase precession was also not caused by differences in LFP power, because LFP power was similar across the stimuli (Supplementary Fig. 2). We also controlled for the possibility of a phase reset evoked by stimulus onset that could differentially affect spike-phase locking for the different stimuli (Supplementary Fig. 11). The absence of an influence of phase-reset was confirmed with a control analysis in

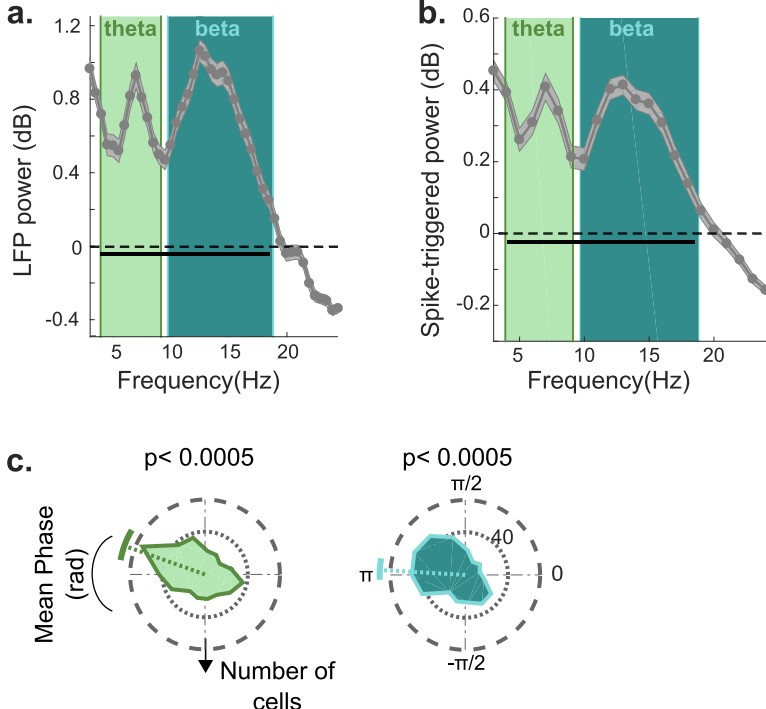

**Fig. 2 LFP power and theta and beta phase-locking of neurons. A** LFP power (expressed in decibel units relative to a $1/f^\alpha$ fit, see Methods) reveals prominent activity in the theta (4−8 Hz) and low-beta (10−18 Hz) bands. **B** Theta and beta oscillations in the power spectra of the spike triggered power (STP) during the sequence learning sessions (relative to a $1/f^\alpha$ fit). Again, peaks are observed in the theta and beta bands. In **A**, **B**, the solid gray lines and shaded areas correspond to the mean and *SEM* across all recording channels, respectively. The solid black horizontal lines represent frequencies in the theta and beta ranges that were significantly different from a $1/f^\alpha$ fit. **C** Distribution of preferred firing phases across cells in the sequence learning sessions with respect to the theta and beta oscillations. The colored dotted and angular solid lines correspond to the mean and standard deviation of the preferred firing phase across cells, respectively. The *p*-values are from a Rayleigh test.

which we removed precisely stimulus-locked spikes around the time of maximum phase reset; again, the phase precession effect remained significant. Next, since theta oscillatory activity in the primate[21] and human hippocampus[22,23] is often observed to be fragmented, with alternating periods of low and high theta power, we asked whether stimulus-specific phase encoding was influenced by variations in theta power. During periods of both low and high theta we found the same pattern of results as before for the preceding and following stimuli: the neurons fired at distinct theta phases, and the phase values shifted forward during the sequence (Supplementary Fig. 12). Finally, we examined how phase precession relates to various factors such as the reliability of stimulus preferences, strength of phase-locking, electrode location, the time course of learning, or the length of the stimulus sequence; the main finding was robust across variations in these factors (Supplementary Fig. 13).

## Discussion

Influential models of the hippocampus posit that this brain area structures incoming information by generating sequentially organized cell assemblies, each for a different input or event. It has been hypothesized that a sequence of spatial locations (during navigation), or a sequence of items that has been learned, is organized at different phases of the theta rhythm, which might organize cell assemblies into sequences via phase precession[12,24]. Our results provide important new insights into phase precession by showing (1) that phase precession occurs in the human brain and (2) occurs outside navigation, in a task that required the learning of a sequence of visual objects. Neuronal firing elicited by different items in a sequence was phase-locked to the theta

oscillation at distinct phases, and we observed a gradual phase precession that spanned three items in the sequence, with spikes fired at successively earlier phases as the sequence progressed. The phase coding was a robust phenomenon, which was also observed in numerous control analyses.

Theta oscillations and phase precession are prominent in the rodent hippocampus during spatial navigation[11]. Spatial navigation in virtual environments in humans has revealed that navigational goals are represented in the firing activity of neurons[25]. Theta oscillations have also been observed in humans, with frequencies ranging from 3 to 9 Hz[10,18,19,26–28]. Watrous et al. (2018)[10] recently reported phase locking of action potentials to lower theta frequencies (3 Hz) during navigation in a virtual environment. However, unlike our study, that study did not report a consistent directionality of theta phase coding with respect to the navigational trajectory of the participant, a difference that might be due to the different theta frequency range, or the different task. Here, we created a "conceptual" space, which consisted of a set of images that were ordered in a sequence and observed a distinct spike-phase code for successive items in the sequence.

Recent studies in human and non-human primates have investigated how a series of items during memory tasks is encoded in different brain regions. At the level of single neurons, it appears that the last presented item is most reliably encoded in firing activity[2,3], but these studies did not report phase locking to ongoing oscillations. Other studies have reported spike-phase coding for higher frequency (32 Hz) oscillations[29], or elevated theta-gamma phase-amplitude coupling[30–32]. An MEG study by Heusser et al.[30], is perhaps most relevant to the current study. They asked participants to learn and recall the order of a

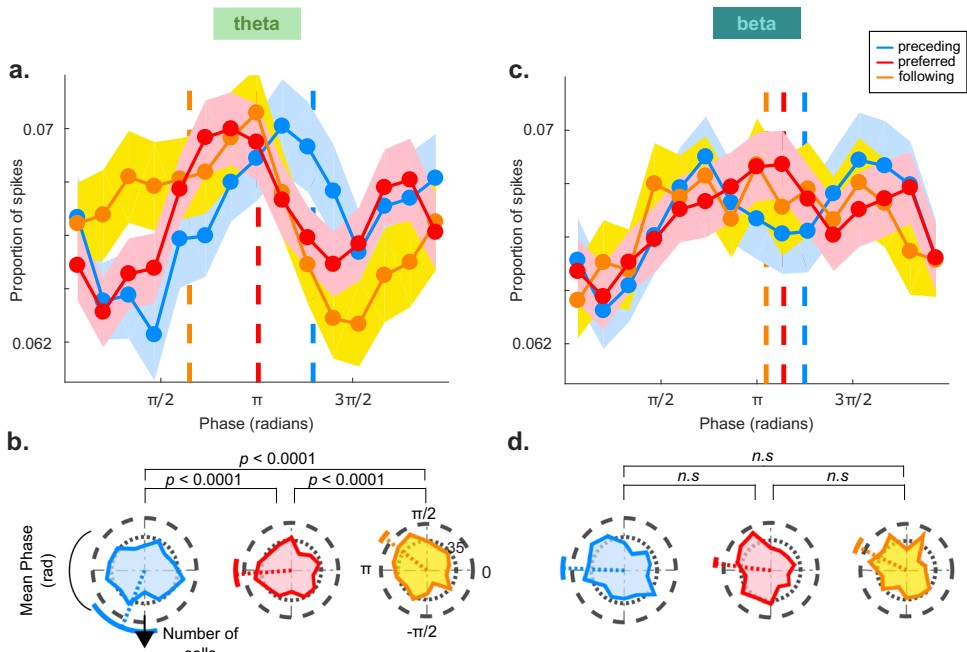

**Fig. 3 Phase encoding of successive stimuli. A** Distribution of spike phases relative to the theta band LFP for the preceding (blue), preferred (red) and following (yellow) stimuli across cells ($N = 452$). The dashed colored lines indicate the mean phase, the shaded areas correspond to the *SEM* across cells. The phase distributions have been smoothed for display purposes, but the mean phase (dashed lines) is calculated for unsmoothed data. **B** Circular histogram of the mean firing phase across all cells ($N = 452$), with colors as in panel (**A**). The dashed colored lines indicate the mean phase, and the colored angular lines correspond to one standard deviation across cells. The p-values above the rose plots are the results of a two-sample Watson-William test for equal means, Bonferroni-corrected for three comparisons. Supplementary Fig. 2 shows the rose plots when pooling spikes across cells. **C**, **D** Spike phase distribution and preferred phase histograms relative to the beta band oscillation for the three stimulus types. In contrast to the theta band, the cells did not fire at distinct phases for the different stimuli. The format of these panels is similar to panels **A**, **B**.

sequence of images and showed that, on successfully encoded trials, consecutive items in the sequence exhibited higher gamma power at distinct theta phases. Our results go beyond by demonstrating theta-phase-coding of action potentials. This is important because models of STM[13] and hippocampal function[12], posit a key role for neuronal spiking activity phase-locked to theta rhythms.

Theta-phase coding models of STM[13] propose that a phase precession-like code could underlie the encoding of a sequence of items (e.g., a phone number). A salient feature of our experimental paradigm is that the task display and instructions evoked the impression of an event sequence, and participants had to learn the temporal order of items. Future studies could investigate if phase precession also occurs for tasks in which participants report about the items in the sequence but in which the order itself is irrelevant.

Phase precession was discovered in the rodent hippocampus. While our report of the advancement of spike phases during sequence-learning bears an overall resemblance to these previous findings, the differences between paradigms raise several questions for future research. For example, rodent place cells show strong preferences for particular spatial positions, and the distance between place fields is reflected in the firing patterns of individual cells[33]. In our task, the cognitive space was a sequence of visual items, which is not strictly analogous to navigation in a two-dimensional space, and cells with strong selectivity for their preferred stimuli showed weaker phase precession (Supplementary Fig. 13G), raising interesting questions for future research about the role of the so-called "concept cells"[34] in the human hippocampal framework[35]. Furthermore, as explained above, in this study we investigated spike-phase coding at the population level and included all cells with a reliable (but sometimes weak)

stimulus preference. Thus, stimulus preference was weaker and firing rates were lower than is typically reported for rodent place cells. In this context, it is worth noting that the strongest phase-coding effects were observed for the full population of responsive neurons; the effects were still visible but statistically weaker when smaller groups of neurons were tested (Supplementary Fig. 13). Therefore, phase-coding in our study is an ensemble effect, more easily detectable in larger populations (although see Supplementary Figs. 3−8 for examples of individual neurons).

Another difference between our results and rodent phase precession is that in the rodent, instantaneous firing rates correlate with spike phase[36], but our results suggest that different phase values are observed without a concomitant difference in firing rate (Supplementary Fig. 10). Finally, it is of interest to compare the rate at which phase precession proceeds between the two species. In rodent place cells, the rate of phase precession is not fixed, but varies as a function of the spatial extent of the place field, and the speed at which the animal crosses it[37]. For example, when the same place field is crossed at slow or fast speeds (within e.g. 12 cycles or 5 cycles), the phase shift from cycle to cycle slows down or speeds up accordingly. In our sequence, the number of stimuli that elicit elevated spiking is one to two stimuli (Fig. 1D)[6]. It took ~24 cycles (2 s/stimulus and two stimuli for a 6 Hz oscillation) to traverse the region in stimulus space associated with increased activity. This is a large number of cycles compared to the number that pass by when a rat traverses a hippocampal place field, and consequently the phase shift from cycle to cycle (or the phase lag reported in Fig. 4) was lower than that reported during spatial navigation in rodents. Future work could test how the rate of phase precession in a conceptual navigation space in humans varies as a function of the presentation speed of the sequence.

Although our paradigm measures the response of single neurons to distinct, successive stimuli, it is of interest to consider the response of multiple neurons (with distinct stimulus selectivity) within a single theta cycle. The finding that it is possible to track the phase advance of one neuron across theta cycles implies that a sequence of items is represented by a set of neurons tuned to different pictures within every theta cycle, firing at specific phases

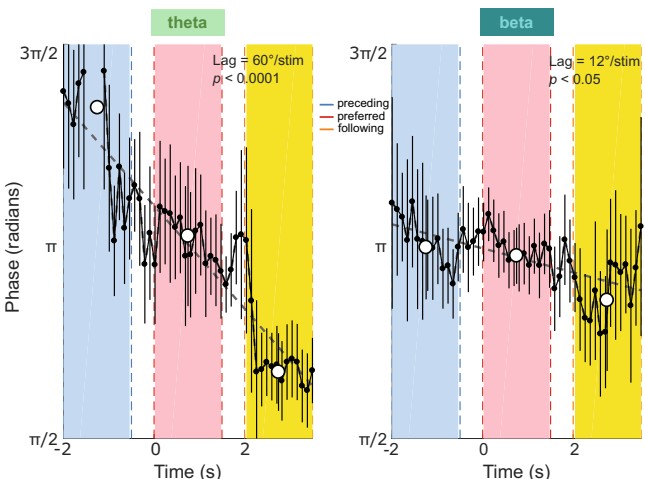

**Fig. 4 Phase as a function of time in the theta (left) and beta (right) bands.** The blue, pink and yellow shaded areas correspond to the stimulus periods of the preceding, preferred and following stimuli. The white regions between stimuli are inter-stimulus intervals. For visualization purposes only, the black dots and lines correspond to the circular mean phase and standard deviation across cells ($N = 452$ cells) computed in overlapping sliding windows of 1.5 s. The circular mean was only plotted when the requirements for confidence limits were met. The large white points correspond to the circular mean phase in non-overlapping time bins centered on the preceding, preferred, and following stimuli. The phase lag value (in units of degrees/stim) reported at the top of each panel is calculated from non-overlapping bins as follows:
$Phase\ Lag = \left(mean_{prec} - mean_{foll}\right)/2$. Significance of the phase lag was determined by a Watson-William test on the phase distributions for the preceding vs. following stimuli. The gray dashed lines correspond to the best linear fit through the data, and is plotted for illustration purposes. The phase lag was significant in the theta band (60°/stimulus, Watson-William test, F(1,903) = 48.3, $p < 0.0001$), and just significant in the beta band (12°/stimulus, Watson-William test, F(1,903) = 5.5, $p < 0.05$).

(Fig. 5). During the cycle, neurons whose preferred stimulus is the previous item fire at the earliest phase and are followed by neurons coding for the current and following stimulus, analogously to the behavior of place cells.

In summary, learning and maintaining the order of a series of stimuli or events is crucial in many tasks. An accurate encoding of a sequence of ordered stimuli enables an organism to predict the future based on regularities learned in the past. Some authors have argued that the role of the hippocampus is to encode events that occur in a temporally organized sequence[38,39], by generating sequentially organized cell assemblies[12,24]. Our results are broadly consistent with these hypotheses in that we observe distinct theta phase firing for successive items in a sequence, that reflects the sequence order. Nevertheless, the importance of phase coding is still under dispute–some species, such as bats, have excellent navigational capabilities and similar neuronal place coding strategies, which do not depend on the theta rhythm[40,41], although more recent results suggest non-oscillatory phase coding and phase precession in the bat hippocampus[42]. The precise role of theta-phase precession therefore remains to be determined. It is encouraging that it is now possible to systematically study theta phase-coding in the human brain so that future studies can also use this approach to test the generality of theta-phase shifts during sequence coding, navigation in real and conceptual spaces, STM, and other cognitive functions.

## Methods

Participants were nine patients (four females, age range 18−36 years) with pharmacologically intractable epilepsy undergoing epileptological evaluation at the Amsterdam University Medical Center, location VUmc, The Netherlands. Patients were implanted with chronic depth electrodes for 7−10 days in order to localize the seizure focus for possible surgical resection[43,44]. All surgeries were performed by J.C.B and S.I. The Medical Ethics Committee at the VU Medical Center approved the studies, and informed consent was obtained from participants. The electrode locations were based entirely on clinical criteria and were evaluated based on the pre-surgical planned trajectories on the basis of structural MRI scans. The accuracy of the implantation was always checked using a CT scan co-registered to the MRI. We only included electrodes that were within a 3 mm deviation from the target (based on visual confirmation).

Each electrode contained eight microwires (Behnke-Fried electrodes, Ad-Tech Medical) from which we recorded single/multi-unit activity and local field potentials, and a ninth microwire that served as a local reference. The signal from the microwires was recorded using a 64-channel Neuralynx system, filtered between 1 and 9000 Hz, sampled at 32 KHz. On average, each patient was implanted with 34 ± 10 microwires (mean ± standard deviation across patients, range = [16,48]). Participants sat in their hospital room at the Epilepsy Monitoring Unit, and performed the experimental sessions on a laptop computer. All patients participated in the two types of experimental sessions described below.

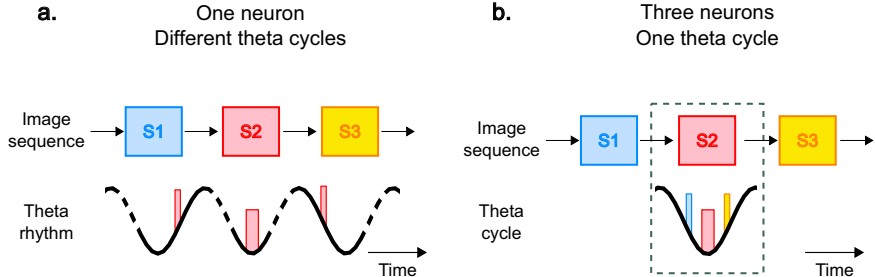

**Fig. 5 Summary of main results. A** A participant views a sequence of three images (S1 to S3). A neuron whose preferred stimulus is S2 fires at the trough of the oscillation during the presentation of S2 (~180°; see Fig. 3B). S1 and S3 are the preceding and following stimuli for this neuron and it fires at phases ~230° and ~120° when S1 and S3 are presented, respectively (Fig. 3B). The thickness of the bars represents the firing rate. The stippled lines for the theta rhythm indicate that the theta cycles are not necessarily consecutive. **B** While stimulus S2 is the preferred stimulus for one neuron, it is also the following stimulus for the neuron selective to S1, and the preceding stimulus for the neuron selective to S3. When S2 is presented on the screen, the neuron whose preferred stimulus is S2 fires around the trough of the oscillation. The neuron whose preferred stimulus is S1 (in blue) will fire at ~120° (S2 is the following stimulus for this neuron). Finally, the neuron whose preferred stimulus is S3 (in yellow) will fire at ~230° (S2 is the preceding stimulus for this neuron). Thus, in a given theta cycle, successive items are represented in the same order as the temporal order of the sequence.

**Screening sessions**. On each day that the patient was available, s/he first performed a screening session during which we presented a large variety of different images (famous people, relatives, animals, landmarks, objects etc.). Each image subtended 2−3 degrees of visual angle and was presented at the center of the screen. Images were presented for 1000−1200 ms, followed by an inter-stimulus interval of 500−700 ms. Each image was repeated 8 times in a randomized order. Between 7 and 51 images were used in the screening sessions depending on the patient's availability. After the presentation of each image the patients performed a simple yes/no task, for example "Did the picture contain a human face"? The exact question depended on the picture set. This task ensured that patients attended to the stimuli. Data from the screening sessions were analyzed to determine which images elicited stimulus-selective responses in some of the recorded neurons[6]. These images from the screening sessions were included in the sequence learning experiment, along with a number of arbitrarily selected images.

**Sequence Learning (SL) Sessions**. The patients performed a total of 29 sequence learning (SL) sessions. In each SL session, participants were presented with a sequence of 5-7 images (image number determined as a function of the difficulty level and the availability of the patient; 1 session with 5 stimuli, 13 sessions with 6 stimuli, and 17 sessions with 7 stimuli). The images were always in a pre-determined order such that a given image, A, predicted the identity of the next image, B, and so on. Participants were asked to remember the order of the stimuli in the sequence. Each stimulus was presented for 1.5 s with an inter-stimulus interval (ISI) of 0.5 s, resulting in individual trials of 2 s (Fig. 1). The sequence was repeated continually 60 times resulting in experimental sessions of ~10−14 min, not including time spent by the participant to respond on probe trials. 20% of trials were "probe" trials in which, instead of being presented with the next image of the sequence, participants were shown two images side by side and asked to decide (by pressing one of two keys on the keyboard) which of the two was the next image in the sequence.

To further the impression of a sequence of images we used the following display arrangement (Fig. 1): Each image was presented at the center of the screen while placeholders (empty gray squares) were presented to the left and right of the central image. At the end of the 1.5 s presentation period, the central image was replaced by a gray placeholder and all the gray squares moved one "step" forward in a clockwise direction for the duration of the ISI, such that each placeholder eventually occupied the next placeholder position. At the end of the ISI, the placeholder that now occupied the central position was replaced by the next image in the sequence. The viewer's subjective impression at the end of the ISI interval was that the central image had been hidden, and then moved clockwise, while the central position was replaced by the next image in the sequence.

**Neuronal stimulus preferences**. As has been previously reported, a subset of human medial temporal lobe neurons (~20% of recorded neurons) show highly sparse and stimulus-selective responses[15]. These "strongly-tuned neurons" are typically identified with screening sessions in which each neuron is tested with a wide set of stimuli, and the stimulus that elicits a large increase in firing activity is identified as the preferred stimulus of the neuron. The remaining recorded cells (~80%) are frequently neglected.

However, recent studies show that weakly-tuned neurons with weak stimulus dependence can also participate in a reliable neural code (e.g., [16,17]). Therefore, in the current study we also wished to investigate population-level dynamics and include cells that might show a weaker preference for a given stimulus, but which may nonetheless be reliable across repeats of the stimulus. To this end, we identified preferred stimuli and verified their reliability using the non-parametric cross-validation procedure described below. This approach allowed us to retain a large number of cells in our population (80%), along with the strongly-tuned neurons, while excluding cells with inconsistent stimulus preferences.

For each neuron we identified the stimulus that elicited the largest number of spikes (i.e., the "hot spot" in the sequence) and designated this stimulus as the preferred stimulus. The stimuli before and after the preferred stimulus were labeled as the preceding and following stimuli, respectively. To verify that the stimulus preference was reliable for each cell, we performed a cross-validation analysis in which we split the data into two halves (and we repeated this process randomly on $10^5$ iterations for each cell). The preferred stimulus was chosen on one half of the data as the stimulus with the largest response. In the second half of the data, we tested the consistency in stimulus preference by verifying that the median response to the preferred stimulus was larger than the median response to the other stimuli (chance = 50%). The consistency measure (reported on subplots (b) of Supplementary Figs. 3-8) was the proportion of iterations on which the preferred stimulus was consistent across the two halves of the dataset. Significance was determined by a binomial test. Only cells ($N = 452$) with a significant consistency score (Binomial test, $p < 0.05$) were included in the final analysis. Of these, 102 (~22.5%) were "strongly-tuned" (i.e., their response covaried significantly with stimulus identity as determined by an ANOVA, $p < 0.05$). The remaining 350 cells were labeled as "weakly-tuned"–cells with weaker but reliable stimulus preferences.

**Spike detection and sorting**. Spike detection and sorting were performed with wave_clus[45]. Briefly, the data were band pass filtered between 300 and 3000 Hz and spikes were detected with an automatic amplitude threshold (Supplementary Fig. 14). Spike sorting was performed with a wavelet transform that extracted the relevant features of the spike waveform. Clustering was performed using a super-paramagnetic clustering algorithm. Clusters were visually reviewed by the first author for (1) the mean spike shape and its variance; (2) the ratio between the spike peak value and the noise level; (3) the inter spike interval distribution of each cluster; (4) the presence of a refractory period for the single-units; i.e. fewer than 1% of spikes in a 3 ms or smaller inter- spike interval; (5) the similarity of each cluster to other clusters from the same microwire. Based on manual inspection of these criteria, clusters were retained, merged or discarded. Unit quality metrics are shown in Supplementary Fig. 15.

**Number of neurons and their locations**. Over the nine patients we recorded from 551 neurons in the left and right hippocampi ($N = 429$) and temporal cortices ($N = 122$) in the sequence learning sessions. Of these, 362 hippocampal neurons and 90 cortical neurons showed significant consistency scores for stimulus preference and were included for further analysis. Single units were manually classified based on the shape and variance of the waveform, the ratio between the waveform peak value and the noise level, the inter-spike interval distribution, and the presence of a refractory period. 126 units were classified as single units.

**Data analysis**. All analyses were performed with the FieldTrip toolbox[46] and custom Matlab code.

*Estimation of theta-band oscillatory activity.* All LFP analyses were performed with the FieldTrip toolbox in Matlab[46]. The LFP was recorded from the same micro-wires as the spiking activity. It was downsampled to a 1,000-Hz sampling rate and notch filtered (between 45−55 Hz and 98−102 Hz) using a second order Butter-worth filter. For each channel, we computed the time-frequency decomposition between 1 and 40 Hz. The time-frequency decomposition was performed with the "mtmfft" method and Hanning tapers over 2 second epochs of the raw LFP trace over the entire duration of the session (i.e., without epoching). We estimated whether significant theta activity was present in the LFP by fitting a $1/f^\alpha$ function to the power spectrum and taking the ratio (in units of decibels) between the actual power spectrum and the $1/f^\alpha$ fit (Fig. 2A). The value for α in the $1/f^\alpha$ fit was $1.98 \pm 0.4$ (mean ± standard deviation across channels), close to the value previously reported in humans[47]. Significant deviation from the $1/f^\alpha$ fit was estimated with a t-test, Bonferroni corrected for multiple comparisons (79 frequencies). In addition to estimating theta power with the raw traces, we also measured the power spectrum of the oscillations around the time of each spike (spike triggered power; Fig. 2B). We extracted a 1 s LFP segment centered on each spike and extracted the frequency spectrum of each segment. The average power spectrum of these LFP traces was estimated by taking the average of the absolute values (the power) of the spectra of all LFP segments[48]. The resultant power spectrum was fitted to a $1/f^\alpha$ function, and the ratio (in decibel units) computed. Significant deviation from the $1/f^\alpha$ fit was estimated with a t-test, Bonferroni corrected for multiple comparisons. Both measures of quantifying oscillatory power revealed prominent theta activity in the 4-8 Hz range and beta activity in the 10−18 Hz range.

*Estimation of spike-LFP phase-locking.* The LFP traces were band-pass filtered with a second order Butterworth filter in the theta (4−8 Hz) and beta (10−18 Hz) frequency ranges. A phase value for each spike in each frequency range was extracted using the Hilbert transform on the band-passed signal, using the Hilbert transform option in the FieldTrip toolbox. The phase value was the angle of the value returned from the Hilbert transform at the time of the spike. A phase of 0° corresponds to the peak of the LFP oscillation, and a phase of ±180° corresponds to the trough of the oscillation. Phase-locking was evaluated by comparing the distribution of phase angles against the uniform distribution using the Rayleigh test. We observed significant phase-locking in the theta and beta ranges (Fig. 2C, Supplementary Fig. 14G). For all analyses, mean phase values were computed as the angle of the mean resultant vector using the Circular Toolbox for Matlab[49].

*Stimulus-specific spike-phase-locking.* To determine whether the spikes for the different stimuli occurred at different phases, we assigned phase values to each stimulus depending on the time at which the spikes were fired. Phase values were binned into 15 bins. For visualization purposes only, a smoothing of two bins in both directions was applied. The angular standard deviation (half of the 68.3% confidence interval) was reported on each plot (again, solely for visualization). To statistically compare the phase distributions between stimuli, the Watson−William test from the Circular Toolbox for Matlab[49] was applied to the unsmoothed phase values.

Only sequence iterations in which the preceding, preferred and following images occurred consecutively were included in the final analyses (Figs. 3 and 4). In other words, sequence iterations in which any of these images corresponded to a probe event were excluded. The mean ± standard deviation of included sequence iterations across cells was $24.1 \pm 4.6$. To control for the effect of spike number on phase between the following and preceding stimuli (Supplementary Fig. 10), we equalized the number of spikes for each neuron by only including the subset of iterations on which the difference between the two stimuli was <=1 spike. We re-

computed the preferred phases for the preceding and following stimuli, while only considering the reduced number of sequence iterations (mean ± standard deviation of included sequence iterations across cells = 14.5 ± 6.9). In the population-level figures (Fig. 4 and Supplementary Fig 13), for display purposes only, the plotted phase values are the circular mean phase values across cells computed in sliding windows (steps of 0.112 s corresponding to 50 steps in each figure). The sliding windows were centered on each time bin, and any window that expanded beyond the trial edges (i.e., −2 s and 3.5 s) were truncated accordingly. Phase precession lags were computed by taking the difference of the mean phase across cells in non-overlapping windows for the extreme stimuli (i.e., the preceding vs. the following stimulus), and statistical significance was determined with a Watson−William test. The window size depended on the exact analysis and the number of cells included in the analysis. In the analysis that pooled over all cells (Fig. 4), the window size was smaller (1.5 s) since there was less variability in the data because of the larger number of cells. In analyses where the data were reduced via a median split (e.g., Supplementary Fig. 13B), it was 2 s; and it was 2.5 s for the remaining analyses. The circular mean was only plotted when the requirements for confidence limits were met as determined by the Circular Statistics Toolbox (please see ref. [49] for details).

*Phase-reset and inter-trial coherence (ITC)*. We controlled for the possible influence of phase-reset of the theta oscillation caused by stimulus onset on measures of phase precession (Supplementary Fig. 11). We computed the inter-trial coherence at each time and frequency point. On each trial of the following, preferred and preceding stimuli we extracted an LFP segment in the time window of [−0.5 1.5] sec (time 0 corresponds to stimulus onset). For each channel, we computed the time-frequency decomposition for 32 logarithmically-spaced frequencies between 1.7 Hz and 98.7 Hz[9]. The time-frequency decomposition was performed with the multitaper method with 2 cycles per frequency on the notch-filtered and down sampled signal, in a time interval of −0.5 s to 1.5 s in steps of 50 ms. The phase and power at each time and frequency point was extracted from a time-frequency transform of the signal. The inter-trial coherence is the absolute value of the average spectrum normalized by its amplitude[50], and varies between zero (no phase-locking) and one (perfect phase-locking). Equal numbers of trials were used for the different stimuli.

**Statistical testing**. We used two-tailed tests unless otherwise specified. All statistical tests performed on circular data were performed with the Circular Toolbox for Matlab[49], applied to the unsmoothed data.

**Reporting summary**. Further information on experimental design is available in the Nature Research Reporting Summary linked to this paper.

## Data availability
Source data for Figs. 2, 3, 4, and Supplementary Figs. 10, 11, 12, 13 are provided with this paper. Additional data are available from the corresponding author upon reasonable request. Source data are provided with this paper.

## Code availability
The analysis used the wave_clus toolbox,the FieldTrip toolbox, and the Circular Statistics toolbox.

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

## Acknowledgements

We are grateful to C.J. Stam, E. van Dellen, L. Douw, P. Ris, S. Claus, D. Velis, and R. Joosten for help with obtaining ethics approval and for technical help with the patients and recordings. This work was supported by grants from the French Agence Nationale de la Recherche (ANR-12-JSH2-0004-01 and ANR AI-REPS–18-CE37-0007-01), the Fyssen foundation, and the Université Paul Sabatier, Toulouse, France (BQR, 2009 and Appel à Projets de Recherche Labellisés, 2013), to L.R., the European Research Council (ERC Consolidator Grant P-Cycles number 614244), the French Agence Nationale de la Recherche (ANR OSCI-DEEP ANR-19-NEUC-0004), and an ANITI (Artificial and Natural Intelligence Toulouse Institute) Research Chair (ANR-19-PI3A-0004) to R.V., the Studienstiftung des Deutschen Volkes (German Academic Scholarship Foundation) to B.Z., the European Union (ERC Grant Agreement n. 339490 "Cortic_al_gorithms" and grant agreements 720270 and 785907 "Human Brain Project SGA1 and SGA2') and the Friends Foundation of the Netherlands Institute for Neuroscience to P.R.R.

## Author contributions

L.R. and P.R.R. designed the study. J.C.B. and S.I. performed the surgeries. M.S., B.Z., M.P., J.K.P., J.C.P., and L.R. collected data. L.R. analyzed the data with input from R.V. L.R. wrote the first version of the manuscript. L.R. and P.R.R. finalized the manuscript. All authors commented on the finalized version of the manuscript.

## Competing interests

The authors declare no competing interests.
