## [Peer Review File · Nature Communications]

REVIEWER COMMENTS

Reviewer #1 (Remarks to the Author):

The paper "Theta-phase dependent neuronal coding during sequence learning in human single neurons" by Reddy, Self et al. reports on microwire recordings from the medial temporal lobe in nine epilepsy patients performing an image sequence learning task. The authors report that neurons respond to their preferred image at a different phase of the ongoing oscillatory activity in the theta band than to the preceding and following stimulus. Specifically, this phase is later for the preceding stimulus and earlier for the following stimulus, which the authors interpret as evidence for phase precession, similar to that observed in rodent place cells as the animal traverses a place field during navigation.

Intracranial in-vivo recordings from the human brain are rare and precious, and recordings of single-neuron activity along with continuous local field potentials (LFP) even more so. The methods used in the study are sound and sophisticated. The paper is very well written and comes from one of the few groups worldwide that is well established in the field of human single-unit recordings. The finding of phase precession as a type of temporal coding (as opposed to rate coding) in humans in a non-navigational sequence learning task in my opinion is sensational and deserves publication at the highest level. But although there is a lot to be liked about this paper, I have a few concerns that should be addressed before publication.

Major points.

(1) Both in the abstract and in the paper, the authors relate their findings to theories and models of short-term memory (STM), or rather working memory (WM). However, it appears crucial to me for the observed phase precession that the subjects have already learned the sequence of the stimuli, and have stored it in long-term memory (LTM). In fact, I would like to encourage the authors to speculate in their Discussion about whether they would also expect to see phase precession in a true STM/WM task, in which a different sequence of images is remembered (i.e., held in WM) in each trial. This was done, e.g., in Ref 2. Since I am a co-author of that study, I can disclose that we did not observe phase precession, although we looked for it. I now regret that we did not report this negative finding, but this is exactly why I would like to see the authors discuss the differential task-specific requirements, as well as implications, for WM and LTM, respectively.

(2) How were the 5-7 stimuli per patient selected? In a previous study from the same group (Reddy et al., Nat Commun, 2015), stimuli were selected from a preceding screening session to elicit responses in one or more cells. Was this the case here also? The description in the Methods ("All patients participated in the two types of experimental sessions described below." However, no second type of experimental sessions is described subsequently.) makes me wonder if that second session may have been a preceding screening session. Or were the stimuli completely arbitrary? And the 5-7 stimuli taken from a constant set of 7 stimuli for all subjects?

This has important implication for the interpretation of the findings. Ref. 10 describes the hippocampus as a sequence generator that uses neurons as content-free pointers to assign them to the items in a sequence (for instance of place fields) as needed. And we know that the neurons thus selected come in a pre-wired preferred sequence (see, e.g., Dragoi & Tonegawa, Philos Trans R Soc Lond B Biol Sci, 2013), and that they are probably selected for this very reason.

It would be important to know whether the neurons reported in this study had already "preferred" their preferred stimulus before the beginning of the task, or whether this preference emerged as subjects learned the order of the stimuli in the presented sequence. One way to address this question would be to monitor the degree of response selectivity over the course of the task, and especially during the first few rounds when the learning of the sequence occurs. Could this selectivity have emerged during learning of the sequence?

(3) Along the lines of my previous point, and regardless of how the stimuli were selected, it would be

interesting to see how response selectivity (“stimulus preference”) relates to phase precession. If the authors separated all of their cells via median split (as in Fig. S5) according to their degree of stimulus preference (averaged over the course of the experiment), which group would more clearly exhibit phase precession - those with high or with low selectivity?

Furthermore, albeit less importantly, it would be interesting to see how phase locking relates to phase precession. If a similar median split were performed based on the strength of theta-phase-locking, which group would show stronger phase precession?

(4) In the Discussion the authors state that the rate of phase precession in rodent place cells varies as a function of the spatial extent of the place field and the speed of the animal. While they propose that presentation speed of a stimulus sequence could be investigated in future work, they are apparently missing out on the opportunity to investigate the “human” counterpart of the spatial extent, namely the length of the presented sequence. Subjects were presented with five, six, or seven sequential stimuli. Were there any differences with respect to the observed phase precession between the neurons corresponding to these three groups?

The authors report the mean phase lag between consecutive stimuli to translate into a time lag of ~50 ms. This is interesting as it is not far from the time lags required for spike-timing dependent plasticity (STDP). How high was the variability of this time lag for a given neuron over the course of the experiment? Regardless of the outcome of the analysis suggested above, it would be nice to see a histogram of the mean time lags per cell, color-coded (e.g., as stacked bars) to indicate whether neurons were presented with a sequence of five, six, or seven stimuli. This could for instance be added as a panel to Figure S2.

(5) In rodents, phase precession has been described for hippocampal place cells and entorhinal grid cells, but to my knowledge not in other brain regions. Nevertheless, 20% of the neurons reported here were in temporal or parahippocampal cortex. Since the human parahippocampal cortex includes parahippocampal, perirhinal, and entorhinal cortex, it would be important to know if the parahippocampal microwires were indeed located so far posterior in the parahippocampal gyrus that they indeed recorded from parahippocampal cortex. But more importantly, it would be interesting to see whether phase precession was stronger in the 80% hippocampal neurons, and whether it was at all present in the 20% cortical neurons (with or without the parahippocampal ones).

Minor points.

(1) line 42: When citing Refs. 1-3, the authors might want to consider citing also Kaminski et al., Nat Neurosci, 2017 and Boran et al., Sci Adv, 2019 as studies showing WM maintenance activity in human single neurons.

(2) line 49: “... as the rat approaches the place fields.” I think this is not correct and should read “... as the rat traverses the place fields”.

(3) line 105: I suggest the authors replace “fixed number of stimuli” with “fixed number of stimuli (5-7)”.

(4) line 106: “parahippocampal gyrus” Please be more specific (see my major point 5).

(5) lines 107/108: Are test trials and probe trials the same? Then please stick to one name.

(6) Why did the authors use a 1/f fit for their spectral power analysis shown in Figure 2? The power law for LFP signals recorded from microwires in humans has been shown to scale with $1/f^2$ (Milstein et al., PLoS ONE, 2009). This should be corrected in Figure 2.

(7) line 147: For better clarity, please replace “mean angle \pm standard deviation” by “mean angle \pm standard deviation across cells”.

(8) line 159: Which test was used here? A t-test against zero? Please explain the number 613 as

degrees of freedom.

(9) line 167: Do mean “phase distribution” instead of “spike distribution”?

(10) line 173: Do you mean “two-sample Watson-William test” instead of “multi-sample Watson-William test”? If these were not pairwise comparisons, please clarify.

(11) line 244: When citing Refs. 23-24, the authors might want to consider citing also Mormann et al., Hippocampus, 2005 as the first studies to show theta-gamma phase-amplitude coupling in humans.

(12) line 326: Please specify the range in numbers of microwires for the patients.

(13) line 364: “619 neurons (single and multi-unit)” How many single units, how many multi-units? What criteria did you use to classify them as SU and MU?

(14) line 392: “Significance was estimated with a t-test” What significance? Significant deviation from the 1/f fit? Where is this shown in Figure 2?

(15) line 400: “We observed significant phase locking in the 4-8 Hz (theta range) and 10-18 Hz (beta range).” Wouldn't it make sense to refer to Figure S6G here?

(16) Figure S1: “firing rate differences between all other stimulus pairs” is confusing since there is a group of “other” stimuli that is not meant here. Perhaps use “firing rate differences between all remaining stimulus pairs”?

(17) Figure S3: “The mean and standard deviation of trials per cell was 13 ± 7 Hz (as opposed to 24 ± 5 Hz for the main analysis; Figure 3).” This does not make sense. The number of trials is not measured in Hz. Also, I thought the trial number for the main analysis was 60 (l.337)? Please clarify.

(18) Figures S4 and S5: Where the p-values of the Watson-Williams tests Bonferroni-corrected?

(19) Figure S6: The authors write in the Methods Section that they used 15 bins for the phase values (l. 407). However, unless I miscounted, panel F shows 18 data points (circles). Please clarify. Also, the caption for G states that “The shaded gray area corresponds to the theta band (4-8Hz)”, but that shaded area seems to be missing.

(20) Throughout the paper, when the authors describe a p-value as ‘Bonferroni-corrected for multiple comparisons’, they should state the number of comparisons. E.g., in the caption of Figure S1B this number should be different from that in the caption of Figure 3B (since “other” is now included).

Typos.

-l.197: “the differences in phase are a robust finding” should be “the differences in phase is a robust finding”.

-l.214: “stimulus specific” should be “stimulus-specific”.

-l.357: “first-author” should be “first author”.

-l.404: “Stimulus specific” should be “Stimulus-specific”.

-l.425: “time frequency transform” should be “time-frequency transform”.

Florian Mormann

Reviewer #2 (Remarks to the Author):

Rodent studies have shown that hippocampal neuronal spiking activities progressively precess against the phase of the local field (LFP) theta oscillation. Here, Reddy et al. investigate this phase code in the human medial temporal lobe (MTL) using a visual memory task with human subjects. By measuring the theta phase of unit spikes across trials preceding and following the preferred stimulus of each neuron, they appear to demonstrate for the first time that the preferred phase of units changes across these items in a manner consistent with the theta phase precession reported in rodent studies. This is a well-written and largely well-analyzed study and the results will be of great interest to the field, as they demonstrate the possibility of a remarkably precise temporal memory code in the hippocampus of humans. My main concern, however, is that the data is not entirely presented in a convincing manner, leaving some doubt about the robustness of the observations. Also, some details of the analyses are confusing and could be explained in more detail.

Major issues

My biggest concern is that raw data is not depicted, and the data that is shown is highly processed and somewhat impenetrable. No raw or filtered LFP, or spike or phase rasters are shown, which raises questions about the quality of the data and the theta phase-precession effect. For such a novel finding, at least a dozen or so examples are warranted. In particular: 1) Spike rasters should show the degree and consistency of spiking responses to preferred stimuli. 2) Theta phase scatterplots should be provided, akin to the plots typically found in rodent studies (e.g. see Terada et al Neuron 2017 for examples from a non-spatial experimental design relevant to the current study). 3) Some sample trials with spiking and simultaneously recorded LFP (raw and theta-band filtered) should be shown to illustrate whether robust theta oscillations are present and how reliably spike phases can be assigned. This is particularly important since Fig S4 suggests that theta power is actually quite low during much of the window used to assign spike theta phase.

A second significant concern is that the neurons show extremely small modulation (even if significant) across stimuli. This raises the question as to the strength of their preference for a given stimulus. Judging from Fig S1, it appears to be only a few percent. The effect size should be reported. It is notable that in their previous study firing rates for the preferred stimuli increased by 5 or more Hz, even after learning, whereas in this study the increase appears to be by at least an order of magnitude less, including in comparison to non-adjacent stimuli. This apparent discrepancy should be addressed. The authors seem to be aware of this issue, as they do perform cross-validation analysis, which is a helpful step. But a bit more needs to be done to provide a fully convincing picture. First, bar graphs are not appropriate for showing firing rate data that are not normally distributed (e.g. Fig S7C); box and whisker plots should be used for these and other firing rate plots. Second, within-cell, rather than across-cell, comparisons should be performed (which also provide more statistical power). Third, as noted above, raster plots and instantaneous firing rates ought to be shown, similar to those in their previous study. Permutation tests or shuffles (or perhaps t-tests if can be justified, as in their previous study) could be used to determine whether such preferences are statistically robust on a single cell level. This point is also important to examine for the phase-locking (Fig 2) and phase-precession (Fig 3) analyses: do individual cells show significant phase-preferences or phase-precession slopes? Currently the authors only examine these across the entire population of 619 cells, rather than for each cell. This also begs the additional question whether the results would remain statistically significant if data is restricted to the individual cells that show a statistically significant increase for the preferred stimulus.

Other comments:

In Fig 1C & Fig. S1. To show the reliability of single-cell preferences, it would be helpful if the authors also present the responses of stimuli preceding and following by 2 items. Also, how were these firing rates normalized? I didn't find this information in the methods.

Some details of the cross-validation method are not well-explained. Presumably these were used to

validate the stimulus preference, but this is unclear from the text. How was the left-out trial chosen? Presumably randomly? And was this presumably performed only once for each unit? How many of the cells showed a consistent stimulus preference on the excluded trial? It may be simpler to split trials in half and demonstrate consistency in stimulus preferences across the two sets of trials.

In Fig 2, It is not clear why only the preceding and preferred trials, but not following trials, were used for the spectral calculations. In these figures, the pre-whitened spectra should also be shown. Some single cell examples of spike-triggered power would also be helpful. For panel 2C, are these the phase-preferences across all stimuli & trials? If so, how are these phase-preferences compatible with the changing phase-preferences shown in other figures? Also as discussed above, please show whether individual cells demonstrate significant phase-preferences.

In Fig 3 and 4, please show phase-histograms for trials preceding & following by 2 items. In particular, it is important to see whether cells continue to show any significant phase-preference beyond the immediately border stimuli. And please indicate whether individual cells show significant phase-precession slopes or correlation coefficients.

How prominent are the autocorrelation lags from the single units used in Fig S2 analysis? Please show some individual examples.

It appears that units in this dataset did not demonstrate the anticipatory responses reported in the authors' previous study. Please discuss this apparent difference.

Minor Points:

Depiction of the electrode locations could be helpful as a supplemental figure.

It could also be useful to calculate cross-correlation lags between units which should also presumably demonstrate lags consistent with the theta phase-offsets (see Skaggs et al Hippocampus 1996).

Harris et al Nature 2002 observed that instantaneous firing rates correlate with the mean theta phase in rodents. The current study (e.g. Fig S3) suggests this relationship does not hold in the existing dataset. This difference should be discussed.

The findings of Eliva et al Cell 2018 are relevant to the discussion of the lack of a theta-phase code in the bat.

REVIEWER COMMENTS

Reviewer #1 (Remarks to the Author):

The paper “Theta-phase dependent neuronal coding during sequence learning in human single neurons” by Reddy, Self et al. reports on microwire recordings from the medial temporal lobe in nine epilepsy patients performing an image sequence learning task. The authors report that neurons respond to their preferred image at a different phase of the ongoing oscillatory activity in the theta band than to the preceding and following stimulus. Specifically, this phase is later for the preceding stimulus and earlier for the following stimulus, which the authors interpret as evidence for phase precession, similar to that observed in rodent place cells as the animal traverses a place field during navigation.

Intracranial in-vivo recordings from the human brain are rare and precious, and recordings of single-neuron activity along with continuous local field potentials (LFP) even more so. The methods used in the study are sound and sophisticated. The paper is very well written and comes from one of the few groups worldwide that is well established in the field of human single-unit recordings. The finding of phase precession as a type of temporal coding (as opposed to rate coding) in humans in a non-navigational sequence learning task in my opinion is sensational and deserves publication at the highest level. But although there is a lot to be liked about this paper, I have a few concerns that should be addressed before publication.

We thank the reviewer for these positive comments and for the other points he raised. We address them below in blue. Changes made in the revised manuscript are highlighted in red in the manuscript.

Major points.

(1) Both in the abstract and in the paper, the authors relate their findings to theories and models of short-term memory (STM), or rather working memory (WM). However, it appears crucial to me for the observed phase precession that the subjects have already learned the sequence of the stimuli, and have stored it in long-term memory (LTM). In fact, I would like to encourage the authors to speculate in their Discussion about whether they would also expect to see phase precession in a true STM/WM task, in which a different sequence of images is remembered (i.e., held in WM) in each trial. This was done, e.g., in Ref 2. Since I am a co-author of that study, I can disclose that we did not observe phase precession, although we looked for it. I now regret that we did not report this negative finding, but this is exactly why I would like to see the authors discuss the differential task-specific requirements, as well as implications, for WM and LTM, respectively.

Theories of WM suggest that it is associated with the reactivation of information stored in LTM (e.g. Cowan, 2001, Oberauer, 2003) so that the reactivation of the next item in the sequence (its entrance into WM) is caused by strengthened connections between the neuronal representations of successive items. The changed connectivity would then be considered to be part of LTM. The difference between the

unpublished findings by the reviewer and our results is intriguing. Phase-precession in rodents has usually been observed after the animals knew the environment, which is also in accordance with the idea that phase precession occurs when sequences can be retrieved from LTM. However, there are other differences between our study and Kornblith et al., including (1) unique trial sequences vs. repetitive sequences, and (2) the task that the subjects performed.

1. Regarding the use of unique trial sequences, it is relevant to consider the MEG study by Heusser et al., (2016) in which subjects were presented with short (~3 minute) and unique sequences of objects. Subjects were tested on the temporal order in which the items were presented. This study reports MEG theta-gamma phase-amplitude coupling, and that items in different sequence positions exhibited higher gamma power at distinct theta phases. To the extent that gamma power can be considered as a proxy for multi-unit activity, this study suggests that phase precession mechanisms can also be observed when unique trial sequences are used.
2. The task itself is another difference between studies. In Kornblith et al., subjects remembered a set of items on each trial, but they did not need to remember the sequence. In our study (and in Heusser et al., 2016), subjects had to remember the order of the items. We therefore suggest that task requirements are important.

We hope that the reviewer will disclose this important unpublished finding in his future work. We do not know how to refer to this unpublished work in the revision and we therefore tried to accommodate the topic by emphasizing the importance of learning the sequence itself (P6):

“Theta-phase coding models of STM ¹³ propose that a phase precession-like code could underlie the encoding of a sequence of items (e.g., a phone number). A salient feature of our experimental paradigm is that the task display and instructions evoked the impression of an event sequence, and subjects had to learn the temporal order of items. Future studies could investigate if phase precession also occurs for tasks in which subjects report about the items in the sequence but in which the order itself is irrelevant”

(2) How were the 5-7 stimuli per patient selected? In a previous study from the same group (Reddy et al., Nat Commun, 2015), stimuli were selected from a preceding screening session to elicit responses in one or more cells. Was this the case here also? The description in the Methods (“All patients participated in the two types of experimental sessions described below.” However, no second type of experimental sessions is described subsequently.) makes me wonder if that second session may have been a preceding screening session. Or were the stimuli completely arbitrary? And the 5-7 stimuli taken from a constant set of 7 stimuli for all subjects?

We thank the reviewer for pointing out this omission in the methods section. Some of the stimuli used in the sequence were indeed selected from prior screening sessions, but as in previous studies (e.g., Quiroga, 2005), this image selection procedure targeted a relatively small number of neurons (typically ~20% of neurons were classified as visually-responsive from prior screening sessions). The remaining stimuli in the sequence were arbitrarily chosen.

The 5-7 stimuli were not taken from a constant set that was used for all subjects. The images in the screening sessions were tailored to each patient’s interests in order to increase our chances of identifying selective neurons and their preferred stimuli.

Information about the screening sessions is now included in the Methods section (P8):

*“**Screening Sessions:** On each day that the patient was available, s/he first performed a screening session during which we presented a large variety of different images (famous people, relatives, animals, landmarks, objects etc.). Each image subtended 2-3 degrees of visual angle and was presented at the center of the screen. Images were presented for 1000-1200ms, followed by an inter-stimulus interval of 500-700ms. Each image was repeated 8 times in a randomized order. Between 7 and 51 images were used in the screening sessions depending on the patient’s availability. After the presentation of each image the patients performed a simple yes/no task, for example “Did the picture contain a human face”? The exact question depended on the picture set. This task ensured that patients attended to the stimuli. Data from the screening sessions were analyzed to determine which images elicited stimulus-selective responses in some of the recorded neurons ⁶. These images from the screening sessions were included in the sequence learning experiment, along with a number of arbitrarily selected images.”*

This has important implication for the interpretation of the findings. Ref. 10 describes the hippocampus as a sequence generator that uses neurons as content-free pointers to assign them to the items in a sequence (for instance of place fields) as needed. And we know that the neurons thus selected come in a pre-wired preferred sequence (see, e.g., Dragoi & Tonegawa, Philos Trans R Soc Lond B Biol Sci, 2013), and that they are probably selected for this very reason.

It would be important to know whether the neurons reported in this study had already “preferred” their preferred stimulus before the beginning of the task, or whether this preference emerged as subjects learned the order of the stimuli in the presented sequence. One way to address this question would be to monitor the degree of response selectivity over the course of the task, and especially during the first few rounds when the learning of the sequence occurs. Could this selectivity have emerged during learning of the sequence?

This is a very interesting point. To address this question, we performed an analysis in which we defined selectivity as the difference in firing response to the preferred stimulus vs. the “worst” stimulus for each neuron. We examined how this “preferred-to-worst selectivity” (hereafter referred to as “selectivity” for simplicity) evolved over the course of the experiment. For each neuron, the sequence length was broken into deciles, where each decile corresponds to ~4 repeats of the sequence (excluding sequence iterations that contained probe events), and the change in selectivity was determined with a linear regression analysis. The change in selectivity as a function of time across all neurons was not significant (Figure R1, left panel; slope = -0.01, $p = 0.3$).

We next reasoned that if stimulus preference does emerge during sequence learning, this effect is more likely to be observed in neurons that do not have a preferred stimulus before the beginning of the task, and less likely in neurons in which stimulus preference is already strong before learning (e.g., the visually-selective neurons determined in the screening sessions). We thus split our cells into two groups: 1) a group of “strongly-tuned” neurons identified with a one-way ANOVA, and 2) the remaining “weakly-tuned” neurons, for which stimulus preference was weaker, but reliable (as verified with a new non-parametric cross-validation procedure, suggested by Reviewer 2, and which we describe in more detail in our response to your point 3 below).

For the “strongly-tuned” cells we observed a decrease in selectivity over the course of the experiment (Figure R1 middle panel; slope = -0.06, $p = 0.05$), consistent with our previous study (Reddy et al., 2015 Nat. Comm.) in which we reported that the responses of selective neurons to their preferred stimuli decrease over the course of the experiment. We previously attributed this effect to a habituation of neuronal responses: each stimulus in our experiment is repeated on 60 trials, and the interval between repeats is very short (i.e., for stimulus sequences of 5-7 stimuli the interval between successive repeats of a given stimulus is 4-6 stimuli).

The selectivity in the second group of “weakly-tuned” neurons did not show a clear trend (Figure R1 right panel; slope = -0.001, $p = 0.9$). Hence, our results do not show conclusively that selectivity emerges during learning. Since these results are inconclusive, we decided to not include them in the paper.

Figure R1: Selectivity (“preferred-to-worst”) as a function of time in the experiment. Stimulus selectivity was defined as the difference in firing response between the preferred stimulus and the least-preferred stimulus. The x-axis corresponds to the sequence number in deciles, where each decile corresponds to ~4 repeats of the stimulus (not including the probe events). Left panel: changes in selectivity across all neurons. Middle panel: changes in selectivity in the “strongly-tuned” group of neurons. Right panel: changes in selectivity in the remaining “weakly-tuned” neurons. The black dots and vertical lines correspond to the mean and SEM across cells.

(3) Along the lines of my previous point, and regardless of how the stimuli were selected, it would be interesting to see how response selectivity (“stimulus preference”) relates to phase precession. If the authors separated all of their cells via median split (as in Fig. S5) according to their degree of stimulus preference (averaged over the course of the experiment), which group would more clearly exhibit phase precession - those with high or with low selectivity?

To address this comment, we split our neurons into groups of “strongly-tuned” and “weakly-tuned” cells. We first define these terms, and then the results of the analysis suggested by the reviewer.

Strongly-tuned and weakly-tuned cells:

Unlike in our previous study (Reddy et al., 2015), in this study we did not only focus on the traditional visually-selective cells that are typically identified with screening sessions. Human single neuron data are rare and valuable, and rather than discard a large proportion of our data simply because the cells did not show extremely sparse and selective responses, we wished to harness the power of our dataset and also include cells with weaker, but nonetheless reliable stimulus preferences. Indeed, recent studies have shown that neurons that are “weakly-tuned” (i.e., with weak stimulus dependence) can nevertheless participate in a reliable neural code (e.g., Insanally et al., 2019, eLife), and we thus also included cells

which demonstrated a more distributed, albeit less selective, neuronal response profile for their preferred stimuli. However, it is important to verify the reliability of the stimulus preference of these neurons, and for this purpose, in the revised manuscript we now perform a new consistency analysis suggested by Reviewer 2.

Consistency analysis:

In this consistency analysis, we split the data into two random halves for each cell (and we repeated this process randomly on 10^5 iterations). The preferred stimulus was chosen on one half as the stimulus with the largest response, and we tested the consistency in stimulus preference in the other half of the data, by verifying that the median response to the preferred stimulus was larger than the median response to the other stimuli (chance is 50%). The consistency score was the proportion of iterations (half-splits) on which the preferred stimulus was consistent across the two halves of the dataset. Significance was determined by a binomial test. In the revised manuscript, only cells ($N=453$) that showed significant consistency ($p<0.05$) in this cross-validation analysis were included in the final analysis. Within this population of cells, we identified a group of “strongly-tuned” neurons with an analysis of variance test (ANOVA) of cell responses to the different stimuli in the sequence, $p<0.05$). 102 “strongly-tuned” cells were thus identified. The remaining 351 cells were labeled as “weakly-tuned” -- cells with weaker but reliable stimulus preferences.

Coming back to the point raised by the reviewer, we tested for phase precession in these groups of strongly-tuned and weakly-tuned cells (Figure R2A). In both groups, the distribution of individual neuron slopes was significantly negative as verified by a t-test against 0 ($p<0.01$ for both groups), consistent with a phase precession effect. The “strongly-tuned” neurons had a shallower slope than the “weakly-tuned” neurons, but this difference was not significant ($t(451) = -0.8$, $p=0.4$) and may have been caused by the larger population of weakly-tuned neurons. The phase-coding effect is a population effect, which is more easily detectable in larger populations.

Figure R2 is included in the revised paper as Figure S8. An explanation of the definition of “strongly-tuned” and “weakly-tuned” neurons, and the consistency analysis is described in the Results and Methods sections. Finally, we refer to the new analysis suggested by the reviewer in the Discussion section.

Results (P3):

“Typically, in the human hippocampus, a relatively small subset of “strongly-tuned” neurons (< 20% of recorded neurons) shows strong and sparse visual selectivity for particular stimuli¹⁵, and the remaining neurons are frequently omitted from further analysis. However, recent work shows that such weakly-tuned neurons can participate in a reliable ensemble neural code¹⁶. In the current study, we also included these neurons. For each cell, we identified the preferred stimulus as the stimulus in the sequence that elicited the largest number of spikes (i.e., the “hot spot” in the sequence, Figure 1C). The reliability of each neuron’s stimulus preference was verified with a cross-validation analysis (see Methods); only cells that showed a consistent stimulus preference were included for further analyses ($N=453$). Of these, 102 (~18%) were “strongly-tuned” (i.e., their response covaried significantly with stimulus identity as determined by an ANOVA, $p < 0.05$), and we refer to the rest as “weakly-tuned”. This approach for neuron selection allowed us to retain a large number of cells in our population (~80%). Figure S3 shows examples of individual cells,

and Figure S8 shows that the main results were replicated across “strongly-tuned” and “weakly-tuned” cells.”

Methods (P8):

“Neuronal stimulus preferences: As has been previously reported, a subset of human medial temporal lobe neurons (~20% of recorded neurons) show highly sparse and stimulus-selective responses¹⁵. These “strongly-tuned neurons” are typically identified with screening sessions in which each neuron is tested with a wide set of stimuli, and the stimulus that elicits a large increase in firing activity is identified as the preferred stimulus of the neuron. The remaining recorded cells (~80%) are frequently neglected.

However, recent studies in animals show that weakly-tuned neurons with weak stimulus dependence can also participate in a reliable neural code (e.g.,¹⁶). Therefore, in the current study we also wished to investigate population-level dynamics and include cells that might show a weaker preference for a given stimulus, but which may nonetheless be reliable across repeats of the stimulus. To this end, we identified preferred stimuli and verified their reliability using the non-parametric cross-validation procedure described below. This approach allowed us to retain a large number of cells in our population (80%), along with the strongly-tuned neurons (~20%), while excluding cells with inconsistent stimulus preferences.

For each neuron we identified the stimulus that elicited the largest number of spikes (i.e., the “hot spot” in the sequence) and designated this stimulus as the preferred stimulus. The stimuli before and after the preferred stimulus were labeled as the preceding and following stimuli, respectively. To verify that the stimulus preference was reliable for each cell, we performed a cross-validation analysis in which we split the data into two halves (and we repeated this process randomly on 10^5 iterations for each cell). The preferred stimulus was chosen on one half of the data as the stimulus with the largest response. In the second half of the data, we tested the consistency in stimulus preference by verifying that the median response to the preferred stimulus was larger than the median response to the other stimuli (chance is 50%). The consistency measure (reported on subplots (b) of Figure S3) was the proportion of iterations on which the preferred stimulus was consistent across the two halves of the dataset. Significance was determined by a binomial test. Only cells (N=453) with a significant consistency score (Binomial test, $p < 0.05$) were included in the final analysis. Of these, 102 (~18%) were “strongly-tuned” (i.e., their response covaried significantly with stimulus identity as determined by an ANOVA, $p < 0.05$). The remaining 351 cells were labeled as “weakly-tuned” -- cells with weaker but reliable stimulus preferences.”

Discussion (P6):

“...it is worth noting that while the phase-coding effects are easy to observe for the full population of responsive neurons, the effects were still visible but statistically weaker when smaller groups of neurons were tested (e.g., strongly-tuned vs. weakly-tuned neurons, Figure S8).”

Furthermore, albeit less importantly, it would be interesting to see how phase locking relates to phase precession. If a similar median split were performed based on the strength of theta-phase-locking, which group would show stronger phase precession?

To address this question, we determined the strength of phase-locking for each cell outside the [preceding, preferred, following] window, because our phase precession effects are observed in this window, and phase precession might be expected to negatively affect phase-locking. Therefore, the strength of phase-locking for each cell was determined based on spikes fired to the preceding-1 and following+1 stimuli (phase values were similar for these two stimuli; Figure S2A). The cells were split according to the median phase-locking strength (Figure R2, included in the paper as Figure S8). In both the low and high phase-locking groups, the distribution of slopes was negative (t-test against 0, $p=0.07$ for the low phase-locking group, $p<0.001$ for the high phase-locking group). The difference in individual neuron slopes between the groups was not significant ($t(451) = -0.7, p=0.5$).

Figure R2: Strength of phase precession examined in different conditions. A) The neurons were divided into a group of cells that were “strongly-tuned” (see Methods) as determined via a one-way ANOVA of the firing response across the stimuli presented in the sequence ($p < 0.05$). The remaining “weakly-tuned” cells were cells that showed a weaker, but reliable response as determined by a non-parametric cross-validation analysis. In both groups, the distribution of slopes was significantly negative as verified by a t-test against 0 ($p < 0.01$ for both groups). The difference in individual neuron slopes between the groups was not significant ($t(451) = -0.8, p=0.4$). B) The neurons were split into two groups according to the median phase locking strength. The phase-locking for each cell was determined outside the [preceding, preferred, following] window, since phase precession effects in this window might be expected to negatively affect phase-locking. Thus, the strength of phase-locking was determined based on spikes fired to the preceding-1 and following+1 stimuli (the phase values for these stimuli were similar, Figure S2). In both the low and high phase-locking groups, the distribution of slopes was negative (t-test against 0, $p=0.07$ for the low phase-locking group, $p < 0.001$ for the high phase-locking group). The difference in individual neuron slopes between the groups was not significant ($t(451) = -0.7, p=0.5$). C) Cells were split according to the length of the stimulus sequence. D) Cells were split based on the location of the electrodes. Phase precession slopes were calculated with a linear regression. The gray dashed lines correspond to the best-fit lines calculated from a linear regression of phase versus time. The black dots and lines correspond to the mean and standard deviation across cells.

(4) In the Discussion the authors state that the rate of phase precession in rodent place cells varies as a function of the spatial extent of the place field and the speed of the animal. While they propose that presentation speed of a stimulus sequence could be investigated in future work, they are apparently

missing out on the opportunity to investigate the “human” counterpart of the spatial extent, namely the length of the presented sequence.

In the “rodent maze/human sequence-learning” metaphor, the length of the sequence is in fact not the human counterpart of the spatial extent of a place field. The length of the sequence is akin to the length of the maze or track along which the animal has to run. In our design, the spatial extent of a place field would be more similar to the width of the “tuning curve” around the preferred stimulus. As we have shown before (Reddy et al., 2015), a cell’s stimulus preference extends from the preferred stimulus to at least the preceding stimulus. The extent of this encoding of neighboring stimuli could potentially correspond to the spatial extent of the place field.

Subjects were presented with five, six, or seven sequential stimuli. Were there any differences with respect to the observed phase precession between the neurons corresponding to these three groups?

We calculated the slope of phase precession between these three groups of neurons as suggested by the reviewer. The data for stimulus sequences of six or seven stimuli is shown in Figure R2 (Figure S8 in the revised paper). There was only one session with a sequence length of five stimuli, with a number of neurons too small ($N=14$) to yield meaningful results (we do not show these data in the figure).

We provide information about the number of sessions with 5, 6 or 7 stimuli in the methods section:

“(…1 session with 5 stimuli, 13 sessions with 6 stimuli, and 17 sessions with 7 stimuli).”

The authors report the mean phase lag between consecutive stimuli to translate into a time lag of ~50 ms. This is interesting as it is not far from the time lags required for spike-timing dependent plasticity (STDP). How high was the variability of this time lag for a given neuron over the course of the experiment? Regardless of the outcome of the analysis suggested above, it would be nice to see a histogram of the mean time lags per cell, color-coded (e.g., as stacked bars) to indicate whether neurons were presented with a sequence of five, six, or seven stimuli. This could for instance be added as a panel to Figure S2.

The average time lag between two consecutive stimuli reported in the paper was actually ~28ms (not ~50ms). The variability of this time lag over the course of the experiment depends slightly over what portion of the experiment the time lag is computed. When we estimate the time lag separately for three separate portions of the experiment for each neuron, the variability (standard deviation) across the three time windows is 22ms (mean across cells). When we estimate the variability over five time windows, the variability is 24ms, and it is 25ms for 10 time windows. This variability is large relative to the mean time lag itself.

For consistency with the rest of the paper, we now report phase lags instead of time lags (the two are related by a constant scaling factor). We now provide the histogram of phase lags between consecutive stimuli, gray scale-coded by sequence length, as requested by the reviewer (Figure R3 below, and Figure S2B, C in the paper). The mean phase lag of this distribution is -19.3 degrees/stimulus. This is smaller than the average value of ~-58 degrees/stimulus derived from the population rose plots in Figure 3, but this difference is expected because of the two different ways that the data are analyzed: The cells that have consistent phase-locking without any phase precession would show near-zero phase lags in the plot below, but may be pointing in any direction in the population average.

Figure R3: Distribution of phase lags between consecutive stimuli in the (A) theta (circular median test against 0 phase lag, $p=0.01$) and (B) beta bands ($p=0.6$). The distribution is gray scale-coded by the length of the sequence (there were 17 sessions with seven stimuli, 13 sessions with six stimuli, and one session with five stimuli). The black arrow indicates the mean phase lag value (-19.3 degrees in the theta band, -3.3 degrees in the beta band). This value in the theta band is smaller than the value obtained at the population level (Figure 3). This is because cells that have consistent phase-locking without any phase precession would show near-zero phase lags in this plot, but may be pointing in any direction in the population rose plots.

This figure is discussed in the Results section (P4):

“This corresponds to an average phase difference between consecutive stimuli of -58 degrees/stimulus. This phase lag, measured at the individual cell level rather than the population level, remains significantly negative (-19.3 degrees/stimulus, circular median test against 0, $p=0.01$; Figure S2B) in the theta band, but not in the beta band (-3.3 degrees/stimulus, $p=0.6$; Figure S2C).”

(5) In rodents, phase precession has been described for hippocampal place cells and entorhinal grid cells, but to my knowledge not in other brain regions. Nevertheless, 20% of the neurons reported here were in temporal or parahippocampal cortex. Since the human parahippocampal cortex includes parahippocampal, perirhinal, and entorhinal cortex, it would be important to know if the parahippocampal microwires were indeed located so far posterior in the parahippocampal gyrus that they indeed recorded from parahippocampal cortex. But more importantly, it would be interesting to see whether phase precession was stronger in the 80% hippocampal neurons, and whether it was at all present in the 20% cortical neurons (with or without the parahippocampal ones).

We note that phase precession has previously been reported in other brain regions in the rodent, such as the basal forebrain (Tingley, 2018) or medial prefrontal cortex (Jones, 2005).

As for the location of the parahippocampal electrodes, the data analyzed for this study were collected several years ago (between 2011 and 2016). For some patients we are not able to access the old medical records to verify the exact electrode locations (although we note that the labels for each electrode were assigned by the clinical team, and electrode locations were verified with a CT scan co-registered to the MRI). Consequently, based on this remark and to avoid ambiguity, we have chosen to exclude the presumed parahippocampal cells in our final analyses. We now only focus on cells that were either in the hippocampus ($N=374$) or in cortex ($N=79$). Phase precession in these two brain regions is shown in Figure R2 (Figure S8 in the paper).

Minor points.

(1) line 42: When citing Refs. 1-3, the authors might want to consider citing also Kaminski et al., Nat Neurosci, 2017 and Boran et al., Sci Adv, 2019 as studies showing WM maintenance activity in human single neurons.

We have included these references.

(2) line 49: "... as the rat approaches the place fields." I think this is not correct and should read "... as the rat traverses the place fields".

This sentence has been corrected.

(3) line 105: I suggest the authors replace "fixed number of stimuli" with "fixed number of stimuli (5-7)".

This change has been made.

(4) line 106: "parahippocampal gyrus" Please be more specific (see my major point 5).

As we mention in response to major point 5, we no longer include parahippocampal cells in the analysis.

(5) lines 107/108: Are test trials and probe trials the same? Then please stick to one name.

These terms correspond to the same event in the experiment, and we now refer to them as "probe events".

(6) Why did the authors use a $1/f$ fit for their spectral power analysis shown in Figure 2? The power law for LFP signals recorded from microwires in humans has been shown to scale with $1/f^2$ (Milstein et al., PLoS ONE, 2009). This should be corrected in Figure 2.

Thanks for raising this point. The notation of $1/f$ was a simplification. We did in fact use $1/f^\alpha$ in the analyses. The fitted value for α was 1.98 ± 0.4 (mean \pm standard deviation across channels), close to the value reported by Milstein et al. We have now improved the Methods sections as follows:

"The value for α in the $1/f^\alpha$ fit was 1.98 ± 0.4 (mean \pm standard deviation across channels), close to the value previously reported in humans ⁴⁶."

(7) line 147: For better clarity, please replace "mean angle \pm standard deviation" by "mean angle \pm standard deviation across cells".

This change has been made.

(8) line 159: Which test was used here? A t-test against zero? Please explain the number 613 as degrees of freedom.

Yes, the test was a t-test against 0. The degrees of freedom is now 452 for a cell population of 453 cells. These changes have been made in the revised manuscript.

(9) line 167: Do mean "phase distribution" instead of "spike distribution"?

Thank you for noting this typo. It has been corrected.

(10) line 173: Do you mean "two-sample Watson-William test" instead of "multi-sample Watson-William test"? If these were not pairwise comparisons, please clarify.

Yes, these were two-sample Watson William tests, and this has been updated in the manuscript.

(11) line 244: When citing Refs. 23-24, the authors might want to consider citing also Mormann et al., Hippocampus, 2005 as the first studies to show theta-gamma phase-amplitude coupling in humans.

Thanks for noting this! We have included this reference.

(12) line 326: Please specify the range in numbers of microwires for the patients.

The range of the number of microwires across patients was [16,48]. This information is now provided in the methods section.

(13) line 364: “619 neurons (single and multi-unit)” How many single units, how many multi-units? What criteria did you use to classify them as SU and MU?

As we mentioned in Major Point 5 we have now reduced the number of neurons included in the analysis because we no longer count the parahippocampal cells. Also, as mentioned in Major Point 3, we now only include cells that show a significant consistency score for stimulus preference. As a result, we now include a total of 453 neurons, of which 126 were single units. Single units were manually classified based on the shape and variance of the waveform, the ratio between the waveform peak value and the noise level, the inter-spike interval distribution, and the presence of a refractory period. This is now mentioned in the methods section.

(14) line 392: “Significance was estimated with a t-test” What significance? Significant deviation from the 1/f fit? Where is this shown in Figure 2?

We now clarify in the Methods section that the t-test was with respect to the 1/f^α fit. The significant frequencies in the theta and beta bands are marked by a black horizontal bar in Figure 2.

(15) line 400: “We observed significant phase locking in the 4-8 Hz (theta range) and 10-18 Hz (beta range).” Wouldn’t it make sense to refer to Figure S6G here?

Yes, this change has been made.

(16) Figure S1: “firing rate differences between all other stimulus pairs” is confusing since there is a group of “other” stimuli that is not meant here. Perhaps use “firing rate differences between all remaining stimulus pairs”?

We now explicitly refer to “pairs of stimuli” in the legend of this figure to avoid confusion on this point.

(17) Figure S3: “The mean and standard deviation of trials per cell was 13 ± 7 Hz (as opposed to 24 ± 5 Hz for the main analysis; Figure 3).” This does not make sense. The number of trials is not measured in Hz. Also, I thought the trial number for the main analysis was 60 (l.337)? Please clarify.

This is indeed a typo and we have removed “Hz”. As for the number of trials included in the analysis, this number is less than 60 in the main analysis because we only included sequence iterations in which the preceding, preferred and following stimuli could be considered consecutively (i.e., we excluded sequence iterations if one of these three stimulus events corresponded to a probe event). In the firing rate control

(old Figure S3, revised Figure S5), this number is further reduced because we only include sequence iterations in which the difference in the number of spikes fired between the two stimuli (preceding, following) was ≤ 1 . We now explain this in the methods section (P10):

“Only sequence iterations in which the preceding, preferred and following images occurred consecutively were included in the final analyses (Figures 3 and 4). In other words, sequence iterations in which any of these images corresponded to a probe event were excluded. The mean \pm standard deviation of included sequence iterations across cells was 24.1 ± 4.6 . To control for the effect of spike number on phase between the following and preceding stimuli (Figure S5), we equalized the number of spikes for each neuron by only including the subset of iterations on which the difference between the two stimuli was ≤ 1 spike. We re-computed the preferred phases for the preceding and following stimuli, while only considering the reduced number of sequence iterations (mean \pm standard deviation of included sequence iterations across cells = 14.5 ± 6.9).”

(18) Figures S4 and S5: Where the p-values of the Watson-Williams tests Bonferroni-corrected?

These tests were uncorrected and we now explicitly mention this in the legends for these figures (now Figures S6 and S7).

(19) Figure S6: The authors write in the Methods Section that they used 15 bins for the phase values (l. 407). However, unless I miscounted, panel F shows 18 data points (circles). Please clarify. Also, the caption for G states that “The shaded gray area corresponds to the theta band (4-8Hz)”, but that shaded area seems to be missing.

Thank you for pointing out these issues. Figure S6F (new Figure S9) has been corrected to have 15 bins, and we have updated panel G to highlight the theta band.

(20) Throughout the paper, when the authors describe a p-value as ‘Bonferroni-corrected for multiple comparisons’, they should state the number of comparisons. E.g., in the caption of Figure S1B this number should be different from that in the caption of Figure 3B (since “other” is now included).

We now provide this information for all tests that were corrected for multiple comparisons.

Typos.

-l.197: “the differences in phase are a robust finding” should be “the differences in phase is a robust finding”.

-l.214: “stimulus specific” should be “stimulus-specific”.

-l.357: “first-author” should be “first author”.

-l.404: “Stimulus specific” should be “Stimulus-specific”.

-l.425: “time frequency transform” should be “time-frequency transform”.

We have corrected these typos. Thank you for pointing them out.

Reviewer #2 (Remarks to the Author):

Rodent studies have shown that hippocampal neuronal spiking activities progressively precess against the phase of the local field (LFP) theta oscillation. Here, Reddy et al. investigate this phase code in the human medial temporal lobe (MTL) using a visual memory task with human subjects. By measuring the theta phase of unit spikes across trials preceding and following the preferred stimulus of each neuron, they appear to demonstrate for the first time that the preferred phase of units changes across these items in a manner consistent with the theta phase precession reported in rodent studies. This is a well-written and largely well-analyzed study and the results will be of great interest to the field, as they demonstrate the possibility of a remarkably precise temporal memory code in the hippocampus of humans. My main concern, however, is that the data is not entirely presented in a convincing manner, leaving some doubt about the robustness of the observations. Also, some details of the analyses are confusing and could be explained in more detail.

We thank the reviewer for these positive comments and for the other points s/he raised. We address them below in blue. Changes made in the revised manuscript are highlighted in red in the manuscript and supplemental information files.

Major issues

Comment 1: My biggest concern is that raw data is not depicted, and the data that is shown is highly processed and somewhat impenetrable. No raw or filtered LFP, or spike or phase rasters are shown, which raises questions about the quality of the data and the theta phase-precession effect. For such a novel finding, at least a dozen or so examples are warranted. In particular: 1) Spike rasters should show the degree and consistency of spiking responses to preferred stimuli. 2) Theta phase scatterplots should be provided, akin to the plots typically found in rodent studies (e.g. see Terada et al Neuron 2017 for examples from a non-spatial experimental design relevant to the current study). 3) Some sample trials with spiking and simultaneously recorded LFP (raw and theta-band filtered) should be shown to illustrate whether robust theta oscillations are present and how reliably spike phases can be assigned. This is particularly important since Fig S4 suggests that theta power is actually quite low during much of the window used to assign spike theta phase.

We now provide 12 individual cell examples that illustrate the spiking responses, theta phase scatter plots, and theta oscillatory activity (Figure R11 at the end of this letter, Figure S3 in the revised manuscript) as suggested by the reviewer. We would like to point out that the color bar in Figure S4 does not reflect theta power, but inter-trial coherence. Theta oscillations could be present to some extent, but with a different phase on different repeats of the trial. We have updated this figure (now Figure S6) to label the color bar with 'Inter-trial coherence' to avoid misinterpretation.

Comment 2: A second significant concern is that the neurons show extremely small modulation (even if significant) across stimuli. This raises the question as to the strength of their preference for a given stimulus. Judging from Fig S1, it appears to be only a few percent. The effect size should be reported. It

is notable that in their previous study firing rates for the preferred stimuli increased by 5 or more Hz, even after learning, whereas in this study the increase appears to be by at least an order of magnitude less, including in comparison to non-adjacent stimuli. This apparent discrepancy should be addressed. The authors seem to be aware of this issue, as they do perform cross-validation analysis, which is a helpful step. But a bit more needs to be done to provide a fully convincing picture.

Discrepancy in the strength of modulation of firing activity:

In our previous paper, the analysis focused on a small group of highly sparse and stimulus-selective neurons for which we had deliberately selected stimuli that elicited strong firing responses (i.e., “preferred stimuli”). Firing rates for these preferred stimuli could increase by 5Hz or more as the reviewer notes. However, because of this approach, and since only a subset (< 20% of recorded neurons) of human medial temporal lobe cells shows such sparse and strongly-tuned visual responses (Quiroga et al., 2005), the analysis in our previous study was restricted to a small subset of cells and their preferred stimuli. When we used a similar criterion for cell selection in the current study (an ANOVA of cell responses to the different stimuli in the sequence, $p < 0.05$), 102 “strongly-tuned” cells were identified. This group of cells exhibited phase precession (see Figure R5 in this letter, S8 in the supplemental section).

In addition, we investigated neuronal phase preferences across cells with weaker tuning. Human single neuron data are rare and valuable, and rather than discard the remaining 80% of cells simply because they did not show extremely sparse and selective responses, we wished to harness the power of our dataset and also include cells with weaker, but nonetheless reliable stimulus preferences. Indeed, recent studies have shown that neurons that are “weakly-tuned” (i.e., with weak stimulus dependence) can nevertheless participate in a reliable neural code (e.g., Insanally et al., 2019, eLife), and we thus also included cells which demonstrated a more distributed, albeit less selective, neuronal response profile for their preferred stimuli.

However, as the reviewer points out, it is important to verify the reliability of the stimulus preference of these neurons. To address this point, in the revised manuscript we now perform the new cross-validation analysis suggested by the Reviewer in Other Comments #2 below.

Reliability of neuronal stimulus preferences: new cross-validation analysis

To study how phase preference evolves as the sequence progresses, we identified a “preferred” stimulus in the sequence for each cell and aligned the other stimuli accordingly. The preferred stimulus was the “hot spot” in the sequence: the stimulus that elicited the maximum firing response in a neuron. To test that the phase effects were reliable with respect to this criterion for stimulus selection, we had previously performed the cross-validation analysis that the reviewer refers to (previous Figure S1). However, following up on the reviewer’s suggestion in “Other Comments #2”, we have replaced that analysis with a new cross-validation analysis to verify the consistency of stimulus preference across trials (and potentially, reject cells for which consistency could not be confirmed).

For each cell we split the data into two random halves, and repeated this procedure for 10^5 iterations. The preferred stimulus was chosen on one half as the stimulus with the largest response, and we tested the consistency in stimulus preference in the other half of the data by verifying that the median response to the preferred stimulus was larger than the median response to the other stimuli (chance is 50%). The consistency measure (reported on subplots (b) of Figure S3, Figure R11 at the end of this letter) was the

proportion of iterations on which the preferred stimulus was consistent across the two halves of the dataset. Significance was determined by a binomial test.

In the revised version of the manuscript, we no longer include all cells in the analysis. Instead, we restrict all analyses to only include cells ($N=453$) that showed a significant consistency score ($p < 0.05$) in the cross-validation analysis. All figures and control analyses have been updated to only include this group of cells.

These details are now presented in the Results and Methods sections:

Results (P3):

“Typically, in the human hippocampus, a relatively small subset of “strongly-tuned” neurons (< 20% of recorded neurons) shows strong and sparse visual selectivity for particular stimuli ¹⁵, and the remaining neurons are frequently omitted from further analysis. However, recent work shows that such weakly-tuned neurons can participate in a reliable ensemble neural code ¹⁶. In the current study, we also included these neurons. For each cell, we identified the preferred stimulus as the stimulus in the sequence that elicited the largest number of spikes (i.e., the “hot spot” in the sequence, Figure 1C). The reliability of each neuron’s stimulus preference was verified with a cross-validation analysis (see Methods); only cells that showed a consistent stimulus preference were included for further analyses ($N=453$). Of these, 102 (~18%) were “strongly-tuned” (i.e., their response covaried significantly with stimulus identity as determined by an ANOVA, $p < 0.05$), and we refer to the rest as “weakly-tuned”. This approach for neuron selection allowed us to retain a large number of cells in our population (~80%). Figure S3 shows examples of individual cells, and Figure S8 shows that the main results were replicated across “strongly-tuned” and “weakly-tuned” cells.”

Methods (P8):

Neuronal stimulus preferences: *As has been previously reported, a subset of human medial temporal lobe neurons (~20% of recorded neurons) show highly sparse and stimulus-selective responses ¹⁵. These “strongly-tuned neurons” are typically identified with screening sessions in which each neuron is tested with a wide set of stimuli, and the stimulus that elicits a large increase in firing activity is identified as the preferred stimulus of the neuron. The remaining recorded cells (~80%) are frequently neglected.*

However, recent studies in animals show that weakly-tuned neurons with weak stimulus dependence can also participate in a reliable neural code (e.g., ¹⁶). Therefore, in the current study we also wished to investigate population-level dynamics and include cells that might show a weaker preference for a given stimulus, but which may nonetheless be reliable across repeats of the stimulus. To this end, we identified preferred stimuli and verified their reliability using the non-parametric cross-validation procedure described below. This approach allowed us to retain a large number of cells in our population (80%), along with the strongly-tuned neurons (~20%), while excluding cells with inconsistent stimulus preferences.

For each neuron we identified the stimulus that elicited the largest number of spikes (i.e., the “hot spot” in the sequence) and designated this stimulus as the preferred stimulus. The stimuli before and after the preferred stimulus were labeled as the preceding and following stimuli, respectively. To verify that the stimulus preference was reliable for each cell, we performed a cross-validation analysis in which we split the data into two halves (and we repeated this process randomly on 10^5 iterations for each cell). The preferred stimulus was chosen on one half of the data as the stimulus with the largest response. In the second half

of the data, we tested the consistency in stimulus preference by verifying that the median response to the preferred stimulus was larger than the median response to the other stimuli (chance is 50%). The consistency measure (reported on subplots (b) of Figure S3) was the proportion of iterations on which the preferred stimulus was consistent across the two halves of the dataset. Significance was determined by a binomial test. Only cells ($N=453$) with a significant consistency score (Binomial test, $p < 0.05$) were included in the final analysis. Of these, 102 (~18%) were “strongly-tuned” (i.e., their response covaried significantly with stimulus identity as determined by an ANOVA, $p < 0.05$). The remaining 351 cells were labeled as “weakly-tuned” -- cells with weaker but reliable stimulus preferences.”

First, bar graphs are not appropriate for showing firing rate data that are not normally distributed (e.g. Fig S8C); box and whisker plots should be used for these and other firing rate plots.

We now show firing rate data with the box and whisker plot format (Figure R4A; Figures 1C and S2A in the revised manuscript).

Figure R4. A) Differences in firing rates and LFP amplitudes do not account for phase precession. Phase, LFP amplitude (square root of LFP power), and firing rates are shown for the preceding-1 (light gray), preceding (blue), preferred (red), following (yellow), and following+1 (dark gray) stimuli, in the theta (left column) and beta (right column) bands. The largest phase difference was observed for the preceding and following stimuli in the theta band even though the firing rates and LFP amplitudes were not different (phase difference between the preceding and following stimuli, Watson-William Test, $F(1,905) = 76.3$; $p < 0.0001$, significant after Bonferonni correction for 10 pairwise comparisons of five stimulus types; firing rate difference, $p=0.8$, paired t-test; LFP amplitude difference, $p=0.8$, paired t-test). The firing rate differences were significant for all pairs of stimuli (all $p < 0.005$ uncorrected) except for preceding-1 vs. following+1, and preceding vs. following. The LFP amplitudes were not significantly different between all pairs of stimuli (all $p > 0.05$, paired t-test) in both frequency bands. In the theta band, the phase differences were significantly different for all pairs of stimuli (Watson-William Test, all $p < 0.005$ uncorrected), except for preceding-1 vs. following+1 ($p=0.04$), preceding-1 vs. following, and preferred vs. following+1 ($p > 0.05$). The vertical bars for the phase plots correspond to the mean \pm one standard deviation across cells. In the box and whisker plots, the central mark indicates the median firing rate across cells, the notch reflects the 95% confidence interval for the median, and the box reaches from the first to the third quartile (interquartile range). In the LFP plots the vertical bars correspond to the mean \pm standard error of the mean across cells. B, C) Distribution of phase lags between consecutive stimuli in the (B) theta (circular median test against 0 phase lag, $p = 0.01$) and (C) beta bands ($p = 0.6$). The distribution is gray scale-coded by the length of the sequence (there were 17 sessions with seven stimuli, 13 sessions with six stimuli, and one session with five stimuli). The black arrow indicates the mean phase lag value (-19.3 degrees in the theta band, -3.3 degrees in the beta band). This value in the theta band is smaller than the value obtained at the population level (Figure 3). This is because cells that have consistent phase-locking

without any phase precession would show near-zero phase lags in this plot, but may be pointing in any direction in the population rose plots.

Second, within-cell, rather than across-cell, comparisons should be performed (which also provide more statistical power).

As described above, the cross-validation analysis for verifying stimulus preferences was performed on a within-cell basis.

Third, as noted above, raster plots and instantaneous firing rates ought to be shown, similar to those in their previous study.

Individual cell examples of raster plots and instantaneous firing rates are shown in Figure S3 (Figure R11 at the end of this letter).

Permutation tests or shuffles (or perhaps t-tests if can be justified, as in their previous study) could be used to determine whether such preferences are statistically robust on a single cell level.

As mentioned above, in all analyses we now only include cells whose stimulus preferences are statistically robust as determined by the non-parametric cross-validation test.

This point is also important to examine for the phase-locking (Fig 2) and phase-precession (Fig 3) analyses: do individual cells show significant phase-preferences or phase-precession slopes? Currently the authors only examine these across the entire population of 619 cells, rather than for each cell.

Individual cell examples are shown in Figure S3 (Figure R11 at the end of this letter). The distribution of phase lags between successive stimuli (which is a good proxy for the slope) is shown in Figure R4B above.

This also begs the additional question whether the results would remain statistically significant if data is restricted to the individual cells that show a statistically significant increase for the preferred stimulus.

As mentioned above, the revised manuscript now only includes cells that show a consistent preference for their preferred stimulus. Robust phase-precession effects are observed in this population of cells. In addition, to identify cells that show a statistically significant increase for their preferred stimuli, we performed a one-way ANOVA on the responses of each cell to the different stimuli ($p < 0.05$; the “strongly-tuned” cells described on pages 15-16). The phase analysis performed on this restricted group of cells ($N=102$) again showed significant phase precession (Figure R5; included in the revised paper as Figure S8).

We discuss this point in the Discussion section (P6):

“..., it is worth noting that while the phase-coding effects are easy to observe for the full population of responsive neurons, the effects were still visible but statistically weaker when smaller groups of neurons were tested (e.g., strongly-tuned vs. weakly-tuned neurons, Figure S8).”

Figure R5: Strength of phase precession examined in different conditions. A) The neurons were divided into a group of cells that were “strongly-tuned” (see Methods) as determined via a one-way ANOVA of the firing response across the stimuli presented in the sequence ($p < 0.05$). The remaining “weakly-tuned” cells were cells that showed a weaker, but reliable response as determined by a non-parametric cross-validation analysis. In both groups, the distribution of slopes was significantly negative as verified by a t-test against 0 ($p < 0.01$ for both groups). The difference in individual neuron slopes between the groups was not significant ($t(451) = -0.8$, $p=0.4$). B) The neurons were split into two groups according to the median phase locking strength. The phase-locking for each cell was determined outside the [preceding, preferred, following] window, since phase precession effects in this window might be expected to negatively affect phase-locking. Thus, the strength of phase-locking was determined based on spikes fired to the preceding-1 and following+1 stimuli (the phase values for these stimuli were similar, Figure S2). In both the low and high phase-locking groups, the distribution of slopes was negative (t-test against 0, $p=0.07$ for the low phase-locking group, $p < 0.001$ for the high phase-locking group). The difference in individual neuron slopes between the groups was not significant ($t(451) = -0.7$, $p=0.5$). C) Cells were split according to the length of the stimulus sequence. D) Cells were split based on the location of the electrodes. Phase precession slopes were calculated with a linear regression. The gray dashed lines correspond to the best-fit lines calculated from a linear regression of phase versus time. The black dots and lines correspond to the mean and standard deviation across cells.

Other comments:

1) In Fig 1C & Fig. S1. To show the reliability of single-cell preferences, it would be helpful if the authors also present the responses of stimuli preceding and following by 2 items. Also, how were these firing rates normalized? I didn’t find this information in the methods.

The response to the stimuli preceding and following by two items is now shown in Figure S2A (Figure R4 above). We have removed the normalized firing rates (Figure 1C) as it was only presented for illustration purposes and all the data analysis was performed on non-normalized data. We have replaced Figure 1C with the box-and-whisker plot of firing rate data.

2) Some details of the cross-validation method are not well-explained. Presumably these were used to validate the stimulus preference, but this is unclear from the text. How was the left-out trial chosen? Presumably randomly? And was this presumably performed only once for each unit? How many of the

cells showed a consistent stimulus preference on the excluded trial? It may be simpler to split trials in half and demonstrate consistency in stimulus preferences across the two sets of trials.

We apologize that this analysis was not clear. The old cross-validation analysis has been replaced by the cross-validation analysis suggested by the reviewer (and which is described above in Point 2).

3) In Fig 2, It is not clear why only the preceding and preferred trials, but not following trials, were used for the spectral calculations. In these figures, the pre-whitened spectra should also be shown. Some single cell examples of spike-triggered power would also be helpful.

This was a mistake in the methods section. The spectral calculations were actually performed on all trials of the experiment. The methods section has been updated accordingly. The pre-whitened spectra and individual cell examples of spike-triggered power are shown in Figure R6 (Figure S1 in the revised manuscript).

Figure R6. A) Pre-whitened LFP power. B) Pre-whitened spike-triggered power (STP). C) Individual cell examples of STP (after subtraction of the $1/f^\alpha$ fit).

For panel 2C, are these the phase-preferences across all stimuli & trials? If so, how are these phase-preferences compatible with the changing phase-preferences shown in other figures? Also, as discussed above, please show whether individual cells demonstrate significant phase-preferences.

Yes, in panel 2C the phase preference is calculated across all stimuli and trials. It is true that phase precession works against our measure of phase-locking, but given that the phase precession does not go through the full 360 deg, the two measures are not incompatible. Nevertheless, one might predict that cells with lower phase-locking might show higher phase precession because they span more of the phase space. This prediction is confirmed by performing a median split based on overall phase-locking strength. The phase-precession slopes calculated in both groups are consistent with our reasoning above -- cells in the low phase-locking group show a steeper phase precession slope. The resulting figure is copied below for your consideration (Figure R7). For this reason, in the revised manuscript we have reported a median

split analysis (as requested by Reviewer 1) based on phase-locking computed outside the phase precession interval ([preceding, preferred, following stimuli]). Cells with a low and high degree of phase-locking groups show phase precession slopes and the difference between slopes is not significant (Figure R5B above).

Finally, individual cell examples of phase preferences are shown in Figure S3 (Figure R11).

Figure R7: This figure shows the slope of phase precession for cells with low and high phase-locking. Different from Fig. R5B, the strength of phase-locking was now computed across all spikes, including the [preceding, preferred, following] window during which phase precession is observed. The slope of phase precession was steeper in the group with lower phase-locking (although this difference was not significant, $p > 0.05$), possibly because cells that show strong phase precession might span more of the phase space. Since this figure suggests that phase precession effects negatively affect phase-locking, in the revised manuscript (Figure R5B above) we compute the strength of phase-locking for each cell outside the [preceding, preferred, following] window. In that case, low and high phase-locking groups show similar phase precession slopes.

In Fig 3 and 4, please show phase-histograms for trials preceding & following by 2 items. In particular, it is important to see whether cells continue to show any significant phase-preference beyond the immediately border stimuli. And please indicate whether individual cells show significant phase-precession slopes or correlation coefficients.

We show the phase data for the trials preceding and following by two items in bar graph format in Figure R4 (Figure S2 in the revised manuscript). There are, however, two reasons for not showing them in the same format as Figures 3 and 4. First, suppose that there are only 5 stimuli in the sequence. The stimulus that is preferred +3 is identical to the stimulus that is preferred -2. Hence, the definition of preceding or succeeding stimuli becomes ambiguous at larger stimulus lags. Second, we only included sequences in which the three stimuli of interest (preceding, preferred, following) occurred consecutively without a probe trial (this constraint is now explained in the Methods section). This condition required us to restrict our analysis to ~24 iterations of the sequence (instead of 60). Applying a similar criterion over 5 stimuli (including preceding-1 and following+1) reduced the number of trials that can be included in the analysis (to ~12 trials), making the results less reliable. Therefore, we have chosen to display the phase preferences in a bar graph format.

Individual cell examples of phase precession are shown in Figure R11 at the end of this letter (Figure S3). The individual-cell-level distribution of phase lags between successive stimuli (which is a good proxy for the slope) is shown in Figure R4B.

How prominent are the autocorrelation lags from the single units used in Fig S2 analysis? Please show some individual examples.

Individual examples of autocorrelations are now shown in Figure R8 (Figure S4 in the revised manuscript).

Figure R8. A. Autocorrelations of spikes and LFPs. A) Representative example of the autocorrelation of a neuron and the LFP from the same recording site. The autocorrelation function was computed for the spikes (smoothed with a 25ms standard deviation Gaussian window) and for the LFP filtered in the theta-band. Peaks in the autocorrelations were detected in a time window of ± 70 ms around the peak of the average LFP auto-correlogram across all cells, which was 178ms (this window has been indicated in grey). Note that for this cell, the first side peak for the spikes (black arrow) occurs at a shorter lag than the first side peak of the LFP autocorrelation function. In other words, the frequency of the oscillation is slightly higher for the spikes, as is expected during phase precession. B) Distribution of the difference between the peak lag of the LFP and the peak lag of the spikes across cells. The mean of the distribution (indicated by the black arrowhead) is shifted significantly to the right of 0 (mean \pm sem = 12.58 ± 3.3 ms; 95% confidence interval of the mean = [4.9 20.2], $t(1,104) = 3.3$, $p < 0.005$, t-test against 0). To increase the reliability of the measure, only cells ($N=105$) that showed a clear theta-band peak in the autocorrelograms for both spikes and LFPs were included (i.e., the difference between the detected peak and the minimum of the autocorrelation in the highlighted time window was greater than a threshold of 0.15). C) Individual cell examples of the autocorrelation of spiking activity and the LFP.

It appears that units in this dataset did not demonstrate the anticipatory responses reported in the authors' previous study. Please discuss this apparent difference.

The population of cells included in this study does in fact show the anticipatory response reported previously (Figure R9, Figure 1D in the revised manuscript).

Figure R9. Experimental Design. A) In the sequence learning experiment, a sequence of 5-7 images was presented to the subjects in a fixed order. Each image was presented for 1.5s followed by an ISI of 0.5s. During the image presentation period (1.5s), an image was presented at the center of the screen, flanked by gray placeholders. During the ISI period, a placeholder replaced the central image, and all the placeholders moved in the clockwise direction. At the end of the ISI period, the central placeholder was replaced by the next image in the sequence.

B) An example of a 5-image sequence. The sequence was repeated 60 times in each experimental session.

C) For each neuron ($N = 453$) we determined the image that elicited the most spikes. This image was designated as the preferred image for that neuron. Stimulus preference was verified with a cross-validation analysis

(see Methods). The images before and after the preferred image in the sequence were designated as the preceding and following images, respectively. The preferred image had the highest firing rate; the responses to the preceding and following images were not significantly different from each other ($p=0.8$ paired t-test). In the box and whisker plots, the central mark indicates the median firing rate across cells, the notch reflects the 95% confidence interval for the median, and the box reaches from the first to the third quartile (interquartile range).

D) Anticipatory responses observed during sequence learning. The mean firing response across all cells ($N=453$) is plotted as a function of time. Time=0 is the onset of the preferred stimulus for each cell, time=-2s is the onset of the preceding stimulus and time=2s is the onset of the following stimulus. The blue, pink and yellow shaded areas correspond to the stimulus periods of the preceding, preferred and following stimuli. The white regions between stimuli are inter-stimulus intervals. Mean firing activity was corrected by subtracting baseline activity before the preceding stimulus (-2.5s to -2s). Note the anticipatory firing prior to the onset of the preferred stimuli, as a gradual increase of activity during the preceding stimulus ⁶.

Minor Points:

Depiction of the electrode locations could be helpful as a supplemental figure.

The data reported in this study were collected several years ago, between 2011 and 2016. Unfortunately, for a number of these patients we have not been able to access the medical records to verify the exact electrode locations. As such we are unable to provide a figure with accurate electrode locations. Note that the labels for each electrode were assigned by the clinical team, and electrode locations were verified with a CT scan co-registered to the MRI.

It could also be useful to calculate cross-correlation lags between units which should also presumably demonstrate lags consistent with the theta phase-offsets (see Skaggs et al Hippocampus 1996).

Unfortunately, the number of electrodes per patients is limited so that we do not have many paired recordings for which simultaneously recorded neurons had clear preferences for successive stimuli. Furthermore, many of our cells have low firing rates. This causes the cross-correlation functions to be sparse and causes the stimulus preference, while reliable for each cell, to be not as sharply defined as a typical hippocampal “place field”. Hence, we can only provide anecdotal evidence. The result of an example cell-pair is shown below (Figure R10). The restricted number of pairs in combination with weaker tuning and the absence of clear peaks in many of the correlation functions made it difficult to extend this analysis to the population level. The theta phase coding is therefore visible at a population-

level (even though some individual cells display sharp phase effects, see Figure S3 in the paper and Figure R11 at the end of this letter).

We now explicitly mention these differences between our results and the rodent work, and specifically with respect to this point, that our findings are overall clearer at the population level (the modified text from the discussion is copied after your next point below).

Figure R10. We computed the cross-correlation for three neurons that had preference for three consecutive stimuli in the sequence (stimuli labeled 1, 2, and 3). The cross-correlation histograms (as in Skaggs et al., 1996, Figure 13B) for the three pairs of cells is shown here. Each bar represents the proportion of times in which a spike from the first cell in the pair was followed (or preceded) by a spike from the second cell at a given latency (lag on the x-axis). The peaks in the histograms are shifted to the right, with more of a shift for the stimulus pair in which the stimuli were further apart (1-3 vs. 1-2 or 2-3).

Harris et al Nature 2002 observed that instantaneous firing rates correlate with the mean theta phase in rodents. The current study (e.g. Fig S3) suggests this relationship does not hold in the existing dataset. This difference should be discussed.

We thank the reviewer for pointing out this reference. We now mention this difference with the rodent literature in the manuscript (P6):

“Phase precession was discovered in the rodent hippocampus. While our report of the advancement of spike phases during sequence-learning bears an overall resemblance to these previous findings, the differences between paradigms raise several questions for future research. For example, rodent place cells show strong preferences for particular spatial positions, and the distance between place fields is reflected in the firing patterns of individual cells ³². In our task, the cognitive space was a sequence of visual items, which is not strictly analogous to navigation in a two dimensional space ^{33,34}. Furthermore, as explained above, in this study we investigated spike-phase coding at the population level and included all cells with a reliable (but sometimes weak) stimulus preference. Thus, stimulus preference was weaker and firing rates were lower than is typically reported for rodent place cells. In this context, it is worth noting that while the phase-coding effects are easy to observe for the full population of responsive neurons, the effects were still visible but statistically weaker when smaller groups of neurons were tested (e.g., strongly-tuned vs. weakly-tuned neurons, Figure S8).

Another difference between our results and rodent phase precession is that in the rodent, instantaneous firing rates correlate with spike phase ³⁵, but our results suggest that different phase values are observed without a concomitant difference in firing rate (Figure S5). Finally, it is of interest to compare the rate at which phase precession proceeds between the two species. In rodent place cells, the rate of phase precession is not fixed, but varies as a function of the spatial extent of the place field, and the speed at which the animal crosses it ³⁶. For example, when the same place field is crossed at slow or fast speeds (within e.g. 12 cycles or 5 cycles), the phase shift from cycle to cycle slows down or speeds up accordingly. In our sequence, the number of stimuli that elicit elevated spiking is one to two stimuli (Figure 1D) ⁶. It took ~24 cycles (2 seconds/stimulus and two stimuli for a 6Hz oscillation) to traverse the

region in stimulus space associated with increased activity. This is a large number of cycles compared to the number that pass by when a rat traverses a hippocampal place field, and consequently the phase shift from cycle to cycle (or the phase slope reported in Figure 4) was lower than that reported during spatial navigation in rodents. Future work could test how the rate of phase precession in a conceptual navigation space in humans varies as a function of the presentation speed of the sequence.”

The findings of Eliva et al Cell 2018 are relevant to the discussion of the lack of a theta-phase code in the bat.

This reference is added to the discussion (P7):

Nevertheless, the importance of phase coding is still under dispute -- some species, such as bats, have excellent navigational capabilities and similar neuronal place coding strategies, which do not depend on the theta rhythm ^{39,40}, although more recent results suggest non-oscillatory phase coding and phase precession in the bat hippocampus ⁴¹.

The following pages show examples of individual cells.

Figure R11. (A-L). Examples of individual cells showing phase-precession. The format is the same for all cell examples.

a) 3-seconds of raw and filtered (4-8Hz) LFP traces in a single example trial. The tick marks correspond to simultaneously recorded spikes.

b) Raster plots and instantaneous firing rates for the preceding (blue), preferred (pink), and following (yellow) stimuli. The lighter areas correspond to the 0.5s inter-stimulus interval (ISI). The darker areas correspond to the 1.5s image presentation window. Image onset occurred at time=0s. For each cell, the preferred stimulus was the one that elicited the largest number of spikes across all repetitions of the sequence, and the consistency of stimulus preference was verified with a cross-validation analysis. The consistency score from the cross-validation analysis is reported in the yellow panel (see the methods section for a description of the cross-validation analysis and the consistency score). For all cells shown here, the consistency score was significantly higher than chance (0.5, Binomial test, $p < 0.001$). The black curves and grey shaded areas correspond to the mean and standard error of the mean of firing activity across trials. For determining stimulus preferences all repetitions of each stimulus were included in the analysis, excluding the probe trials. Since the probe events occurred randomly, the number of non-probe trials for each stimulus can be different, resulting in different numbers of trials in the raster plots.

c) Phase distribution of spikes in the 4-8Hz theta band for this cell. The significance was determined by a Rayleigh test.

d) Scatter plot of theta phase versus time. For visualization purposes the phase values are copied vertically three times.

e) Heat map corresponding to the data shown in panel d). For visualization purposes the data are copied vertically three times.

f) Change in phase for this cell as a function of time. The phase lag value reported on the plot corresponds to the phase lag between successive stimuli, in units of radians/stimulus.

In panels c, d, and e), time = 0s is the start of the preferred stimulus, time = -2s is the onset of the preceding stimulus, and time=2s is the onset of the following stimulus. Stimulus onsets are indicated by the thicker vertical lines in panels e) and f), and the ISI onsets are indicated by the thinner vertical lines. For all cells shown here the slopes were significant (regression analysis, $p < 0.05$).

Figure R11 continued.

Figure R11 continued.

Figure R11 continued.

Figure R11 continued.

Figure R11 continued.

Figure R11 continued.

REVIEWER COMMENTS

Reviewer #1 (Remarks to the Author):

The authors have adequately addressed most of my points and have improved and strengthened the manuscript. I do, however, have a few more points that I feel should be addressed prior to publication.

Major Points.

(1) I thank the authors for providing Figure S8, which contains important control analyses. As the authors correctly state in their rebuttal of my previous point 3, the lack of a significant difference between the slope of the strongly tuned vs. weakly tuned neurons (Panel A) may have been caused by the larger population of weakly-tuned neurons. I therefore recommend performing a median split based on the p-value of the ANOVA to separate the 50% more strongly tuned units from the 50% more weakly tuned units.

Since interneurons, which are known to produce less selective, but graded responses to stimuli would also show up strongly in the ANOVA, a further panel should be added that uses a median split of the 453 cells based on a stimulus selectivity score defined as “preferred-to-rest” (or d') in order to separate highly selective from less selective cells.

This is an important point because according to rodent hippocampal theory (Ref. 12), the hippocampus acts as a sequence generator that uses neurons as content-free pointers to assign them to the items in a sequence as needed. These neurons come in a pre-wired preferred sequence and are assigned accordingly. If neurons had a pre-existing content, i.e., a preferred stimulus as seen in concept cells, this would not work, since it could no longer be assigned to a stimulus as needed. I would therefore expect that highly selective neurons show less phase precession than weakly selective ones. Regardless of the results, these two new panels for Figure S8 will in no way diminish the importance of the general findings of this study, but may add valuable insights.

(2) As yet another panel to be added to Figure S8 (i.e. the sixth), I would very much like to see a median split between the first and second half of trials in order to judge whether phase precession is present from the beginning of the experiment or if it evolves over its course.

Minor Points.

(1) With respect to my previous major point 1, I believe it is important to emphasize that the current experiment was not a working memory (WM) task with single-trial learning. I therefore recommend replacing the phrase “subjects learned sequences of pictures” in the abstract by “subjects learned a fixed sequence of pictures”.

(2) On both occasions when the authors cite Ref [16] to make the point that weakly tuned neurons participate in a reliable neural code, they should consider citing also Reber et al., 2019 (PLoS Biol 17:e3000290), which has shown the same not in rodent recordings, but in recordings from humans similar to the ones used in this study.

(3) Results (P3). “Of these, 102 (18%) were...” 102 of 453 is 22.5%, not 18%.

Florian Mormann

Reviewer #2 (Remarks to the Author):

The revised manuscript is much improved. I particularly appreciate that in the supplementary materials for this revision, the authors have now provided some of the raw data for the paper. The new figures, particularly Supp Fig S3, go a long way in helping readers and reviewers understand the quality and features of the data. However, I was surprised by how the spike-theta phase relationships in the sample units were generally weak, particularly in the provided scatterplots. These raise further questions about how they can be reconciled with the provided phase-position slopes in the main figures. Overall, there are some important questions regarding the derivation and reliability of the phase-values central to this study.

--The a panels in Figure S3 examples show filtered and raw traces. The raw traces generally don't appear to have a strong theta component. Apparently the gain is increased on the filtered version (by how much? this should be reported), which makes it difficult to gauge the correspondence. The unfiltered data should ideally be shown at a similar amplification. Also, please provide the phases assigned to the shown spikes. It's hard to see any evidence for phase-precession at the single trial level here.

--Critically, it is difficult to see the correspondence between panels d, e, and f. The scatterplots in d appear generally uniform. There seems to be some heavy smoothing applied to e. How was this smoothing performed? Since Spike Counts are reported, how large are the bins corresponding to these values? The scalebar does not seem to accurately reflect the values apparent by eye (i.e. empty bins show non-zero values, etc.).

--Most concerning is the lack of apparent correspondence with panel f. The authors don't report how the "phase" in these plots was derived. Presumably it is the mean phase? And presumably it was derived from circular statistics? For phase data, the phase should correspond to the angle of the mean resultant vector. I would like to assume that this is how the authors calculated it, but I could not find this specified anywhere. Here again it looks like some kind of smoothing was applied, but details for the smoothing were not provided. In any case the phase-slopes should be provided for raw unsmoothed data. In some examples (e.g. cell C), the smoothed slope in panel f appears to diverge from what can be seen by eye for the density plots in panel e, raising further questions about how the phase values were derived for each time bin. It would be helpful to see how the mean calculated corresponds to the phase histograms from the individual bins, and if f and e were overlaid. It appears that the panel f values were passed into Fig. 4 and Fig S8 analysis, so this is particularly important.

--It's unclear how the mean phase was calculated for Figure 3 as well. It should presumably be the angle corresponding to the mean resultant vector. Obviously, a statistic appropriate to circular data should be used, but I could not find this explicitly stated in the text. I am not sure why smoothing needed to be applied here. Given the large amount of spiking data from a large number of cells, I would assume that the data would already be fairly smooth. The unsmoothed data should also be provided here (and in related supp figures), in light of the questions already raised.

Reviewer #1 (Remarks to the Author):

The authors have adequately addressed most of my points and have improved and strengthened the manuscript. I do, however, have a few more points that I feel should be addressed prior to publication.

Major Points.

(1) I thank the authors for providing Figure S8, which contains important control analyses. As the authors correctly state in their rebuttal of my previous point 3, the lack of a significant difference between the slope of the strongly tuned vs. weakly tuned neurons (Panel A) may have been caused by the larger population of weakly-tuned neurons. I therefore recommend performing a median split based on the p-value of the ANOVA to separate the 50% more strongly tuned units from the 50% more weakly tuned units.

Since interneurons, which are known to produce less selective, but graded responses to stimuli would also show up strongly in the ANOVA, a further panel should be added that uses a median split of the 453 cells based on a stimulus selectivity score defined as “preferred-to-rest” (or d') in order to separate highly selective from less selective cells. This is an important point because according to rodent hippocampal theory (Ref. 12), the hippocampus acts as a sequence generator that uses neurons as content-free pointers to assign them to the items in a sequence as needed. These neurons come in a pre-wired preferred sequence and are assigned accordingly. If neurons had a pre-existing content, i.e., a preferred stimulus as seen in concept cells, this would not work, since it could no longer be assigned to a stimulus as needed. I would therefore expect that highly selective neurons show less phase precession than weakly selective ones. Regardless of the results, these two new panels for Figure S8 will in no way diminish the importance of the general findings of this study, but may add valuable insights.

(2) As yet another panel to be added to Figure S8 (i.e. the sixth), I would very much like to see a median split between the first and second half of trials in order to judge whether phase precession is present from the beginning of the experiment or if it evolves over its course.

These panels (from both the points above) have been added to Figure S8 (pasted below for your convenience). As you correctly point out, the control of separating the highly selective neurons from less selective cells (or “exclusive” selectivity vs. “graded” selectivity) is an interesting one (Figure S8G). To identify cells that showed a strong selectivity for their preferred stimulus vs. the other stimuli, we defined a new selectivity measure that was defined as (firing rate to the preferred/ firing rate to the other stimuli). Cells with “exclusive” selectivity such as concept cells should have a low mean response to the “other” stimuli (compared to the “preferred” response), and therefore higher selectivity according to this measure. For the most selective cells according to this new measure, the phase precession effect was indeed weaker, as you expected.

We discuss this point on P6 of the revised manuscript:

“...and cells with strong selectivity for their preferred stimuli showed weaker phase precession (Figure S8G), raising interesting questions for future research about the role of the so-called “concept cells”³⁴ in the human hippocampal framework³⁵”.

Minor Points.

(1) With respect to my previous major point 1, I believe it is important to emphasize that the current experiment was not a working memory (WM) task with single-trial learning. I therefore recommend replacing the phrase “subjects learned sequences of pictures” in the abstract by “subjects learned a fixed sequence of pictures”.

This change has been made.

(2) On both occasions when the authors cite Ref [16] to make the point that weakly tuned neurons participate in a reliable neural code, they should consider citing also Reber et al., 2019 (PLoS Biol 17:e3000290), which has shown the same not in rodent recordings, but in recordings from humans similar to the ones used in this study.

Thank you for pointing out this missing reference. It has been added on both occasions you refer to.

(3) Results (P3). “Of these, 102 (18%) were...” 102 of 453 is 22.5%, not 18%.

Thank you for pointing this out. We have made the correction.

(Next page): Figure S8. Strength of phase precession examined in different conditions. In all panels, the points correspond to the circular mean of phase values across cells, plotted as a function of time (Methods). The circular mean was only plotted when the requirements for confidence limits were met, and likewise, the slopes were calculated using only these points (the others being treated as missing values). The mean phase for each point was calculated over a sliding time window (see Methods). Phase precession slopes were calculated with a linear regression. The gray dashed lines correspond to the best-fit lines calculated from a linear regression of phase versus time. The black dots and lines correspond to the circular mean and standard deviation across cells.

A) The neurons were divided into a group of cells that were “strongly-tuned” (see Methods) as determined via a one-way ANOVA of the firing response across the stimuli presented in the sequence ($p < 0.05$). The remaining “weakly-tuned” cells were cells with a weaker, but reliable response as determined by a non-parametric cross-validation analysis. In both groups, the distribution of slopes was significantly negative as verified by a t-test ($p < 0.01$ for both groups). The difference in individual neuron slopes between the groups was not significant ($t(450) = -0.8$, $p=0.4$). B) The neurons were split into two groups according to the median phase locking strength. The phase-locking for each cell was determined outside the [preceding, preferred, following] window, since phase precession effects in this window might be expected to negatively affect phase-locking. Thus, the strength of phase-locking was determined based on spikes fired to the preceding-1 and following+1 stimuli (the phase values for these stimuli were similar, Figure S2). In both the low and high phase-locking groups, the distribution of slopes was negative (t-test against 0, $p=0.07$ for the low phase-locking group, $p < 0.001$ for the high phase-locking group). The difference in individual neuron slopes between the groups was not significant ($t(450) = -0.7$, $p=0.5$). C) Cells were split according to the length of the stimulus sequence. D) Cells were split based on the location of the electrodes. E) Cells were split into two groups according to the median F-value from a one-way ANOVA of the firing response across all stimuli presented in the sequence. F) The data for each cell was split according to the first half or second half of trials (early learning vs. later learning). The difference was not significant between the two groups ($p=0.3$). G) Stimulus selectivity for each cell was determined by taking the ratio of the mean response to the “preferred” stimulus vs. the mean response to the “other” stimuli. Cells with strong selectivity for their preferred stimulus, such as concept cells (Quiroga, 2012) have a low mean response to the other stimuli and therefore higher selectivity. Cells were split into two groups according to the median value of this measure of stimulus selectivity. The phase precession slope tended to be weaker for the more selective cells, but the difference was not significant between the two groups ($p=0.6$).

Reviewer #2 (Remarks to the Author):

The revised manuscript is much improved. I particularly appreciate that in the supplementary materials for this revision, the authors have now provided some of the raw data for the paper. The new figures, particularly Supp Fig S3, go a long way in helping readers and reviewers understand the quality and features of the data. However, I was surprised by how the spike-theta phase relationships in the sample units were generally weak, particularly in the provided scatterplots. These raise further questions about how they can be reconciled with the provided phase-position slopes in the main figures. Overall, there are some important questions regarding the derivation and reliability of the phase-values central to this study.

-- The a panels in Figure S3 examples show filtered and raw traces. The raw traces generally don't appear to have a strong theta component. Apparently the gain is increased on the filtered version (by how much? this should be reported), which makes it difficult to gauge the correspondence. The unfiltered data should ideally be shown at a similar amplification. Also, please provide the phases assigned to the shown spikes. It's hard to see any evidence for phase-precession at the single trial level here.

Indeed, as the spectral analysis in Figure S1 shows, theta power is a small fraction of the total power, in part because of the strong $1/f$ trend (however, theta-band power is still significant, as Figure 2A of the main paper reveals). To compensate for this difference in scale, in the a) panels of Figure S3 the scales of the theta-band vs. raw signals had been adapted for visibility purposes, and the scaling value was not reported because it was different for each cell. We have improved the presentation by plotting both signals with the same y-scale (new Figure S3 below).

Given the relatively weak phase-locking, it is difficult to display phase values for each spike. Indeed, we emphasize that the phase precession effect in our study is visible at the population level, but less clear at the single cell or single trial level. This is an important difference from the rodent literature, and this point, and possible reasons for it, had been discussed in our previous revision (pages 6-7). One important reason for this difference could arise from the observation in rodents that the rate of phase precession varies with the speed of the animal and/or the spatial extent of the place field. The rate of phase precession in our study is lower than that typically reported in rodents, and this factor could contribute to making the effect less obvious at the single trial level (e.g., in the 3s trace plotted in panel a). Various other factors could also be important, such as the different experimental paradigms (spatial navigation vs. a conceptual space of images), duration of the experiment/quantity of data (our experiments were always <15 minutes because of patient constraints), differences in firing preferences and firing rates (place cells vs. visual neurons) etc. These differences will be important to address in future work to fully understand the similarities and differences between the rodent and human frameworks.

We have clarified the differences between our study and the rodent literature in the discussion on page 6:

"In this context, it is worth noting that the strongest phase-coding effects were observed for the full population of responsive neurons; the effects were still visible but statistically weaker when smaller groups of neurons were tested (Figure S8). Therefore, phase-coding in our study is an ensemble effect, more easily detectable in larger populations (although see Figure S3 for examples of individual neurons)."

-- Critically, it is difficult to see the correspondence between panels d, e, and f. The scatterplots in d appear generally uniform. There seems to be some heavy smoothing applied to e. How was this smoothing performed? Since Spike Counts are reported, how large are the bins corresponding to these values? The scalebar does not seem to accurately reflect the values apparent by eye (i.e. empty bins show non-zero values, etc.).

The appearance of the scatterplots in sub-panels (d) depends on the spike density: given a fixed point size for representing each spike, very sparse plots (low firing rates, like the example cell in panel G) look rather empty, while dense plots (high firing rates, like the example cell in panel E) look very dense and dark. In both cases, phase precession effects will be difficult to discern—yet we included these panels because they are a “standard” form of display. We also present the “smoothed” version of the same data in sub-panels (e). The data were binned into 12 bins in the x and y directions (i.e., with a bin size of 0.45s in the x-direction, and ~0.5 radians in the y-direction), smoothed with a 2D Gaussian kernel of 1-bin standard deviation, and upsampled by a factor of 3 (bicubic interpolation method) for display purposes. The scalebar represents the expected firing rate (in Hz) at each point.

-- Most concerning is the lack of apparent correspondence with panel f. The authors don't report how the “phase” in these plots was derived. Presumably it is the mean phase? And presumably it was derived from circular statistics? For phase data, the phase should correspond to the angle of the mean resultant vector. I would like to assume that this is how the authors calculated it, but I could not find this specified anywhere. Here again it looks like some kind of smoothing was applied, but details for the smoothing were not provided. In any case the phase-slopes should be provided for raw unsmoothed data. In some examples (e.g. cell C), the smoothed slope in panel f appears to diverge from what can be seen by eye for the density plots in panel e, raising further questions about how the phase values were derived for each time bin. It would be helpful to see how the mean calculated corresponds to the phase histograms from the individual bins, and if f and e were overlaid. It appears that the panel f values were passed into Fig. 4 and Fig S8 analysis, so this is particularly important.

To illustrate the correspondence with panel f) we now have overlaid the curve in panel f) onto panels d) and e). These panels are shown below in the new Figure S3.

Throughout the paper, the phase values for each spike correspond to the angle of the output of the Hilbert transform, at the time of the spike, following band-pass filtering. We had previously provided information about the Hilbert transform, and have now further updated the Methods section in the new revision. The mean phase value (e.g. across spikes in a time window) was derived from circular statistics throughout the paper. The following clarifications have been added to the revised manuscript:

Methods, page 10:

“A phase value for each spike in each frequency range was extracted using the Hilbert transform on the band-passed signal, using the Hilbert transform option in the Fieldtrip Toolbox. The phase value was the angle of the value returned from the Hilbert transform at the time of the spike.... For all analyses, mean phase values were computed as the angle of the mean resultant vector using the Circular Statistics Toolbox”

The individual phase slopes values (in rad/stim) reported on the top of panels f) were indeed calculated from raw, unsmoothed data. The preceding, preferred and following phase values are the (circular) mean phase values for all spikes fired for these stimuli and phase lag values are the circular mean of the difference in phase between successive stimuli. For plotting purposes only (i.e., for the phase vs. time curves in panels f), the circular mean phase was computed in sliding windows of 2.5s (shifted by steps of 0.112s corresponding to 50 steps in each figure). These points are now explained in the legend of Figure S3.

Figure S3 legend:

“f) Change in phase for this cell as a function of time. The phase lag value reported at the top of the plot is calculated from raw, unsmoothed data, and is the circular mean phase lag between successive stimuli, in units of radians/stimulus. This phase lag value was computed by taking the circular mean of the (difference between the mean preceding phase and the mean preferred phase) and the (difference between the mean preferred phase and the mean following phase). For display purposes only, the phase value that is plotted is the circular mean phase computed in sliding windows of 2.5s (shifted by steps of 0.112s corresponding to 50 steps in each figure).”

Finally, as mentioned above, the curve from panel f) has now been overlaid on panel e). We believe that the correspondence between the two panels (e.g., for Cell C) is clearer in this format.

-- It's unclear how the mean phase was calculated for Figure 3 as well. It should presumably be the angle corresponding to the mean resultant vector. Obviously, a statistic appropriate to circular data should be used, but I could not find this explicitly stated in the text. I am not sure why smoothing needed to be applied here. Given the large amount of spiking data from a large number of cells, I would assume that the data would already be fairly smooth. The unsmoothed data should also be provided here (and in related supp figures), in light of the questions already raised.

Thanks for pointing this out. We have now improved the methods section accordingly. The phase value was the angle of the Hilbert transform at the time of the spike, and the mean phase values were obtained from circular averaging (angle of the mean resultant vector). As mentioned in the methods section (page 10), statistical tests were performed with the Circular Statistics toolbox. For Figure 3, the mean phases and statistics were computed for unsmoothed data. As indicated in the legend, smoothing was only applied for visualization purposes, and only for panels A and C. As requested by the reviewer, we now present the unsmoothed histograms corresponding to these panels as a supplemental figure (Figure S2B, C, copied below for your convenience). The rose plots in panels B and D of Figure 3 were unsmoothed, as were the phase data for the different stimulus types presented in the supplemental section.

B, C) Unsmoothed distributions of spike phases relative to the theta and beta band LFPs for the preceding (blue), preferred (red), and following (yellow) stimuli. The format of these panels is the same as Figure 3 A, C). The dashed lines correspond to a sinusoidal fit.

The following pages show examples of individual cells.

Figure S3. (A-L). Examples of individual cells showing phase-precession. The format is the same for all cell examples.

a) 3-seconds of raw and filtered (4-8Hz) LFP traces in a single example trial. The tick marks correspond to simultaneously recorded spikes.

b) Raster plots and instantaneous firing rates for the preceding (blue), preferred (pink), and following (yellow) stimuli. The lighter areas correspond to the 0.5s inter-stimulus interval (ISI). The darker areas correspond to the 1.5s image presentation window. Image onset occurred at time=0s. For each cell, the preferred stimulus was the one that elicited the largest number of spikes across all repetitions of the sequence, and the consistency of stimulus preference was verified with a cross-validation analysis. The consistency score from the cross-validation analysis is reported in the yellow panel (see the methods section for a description of the cross-validation analysis and the consistency score). For all cells shown here, the consistency score was significantly higher than chance (0.5, Binomial test, $p < 0.001$). The black curves and grey shaded areas correspond to the mean and standard error of the mean of firing activity across trials. For determining stimulus preferences all repetitions of each stimulus were included in the analysis, excluding the probe trials. Since the probe events occurred randomly, the number of non-probe trials for each stimulus can be different, resulting in different numbers of trials in the raster plots.

c) Phase distribution of spikes in the 4-8Hz theta band for this cell. The significance was determined by a Rayleigh test.

d) Scatter plot of theta phase versus time. For visualization purposes the phase values are copied vertically three times. The red line corresponds to the mean phase as a function of time as explained in panel f).

e) Heat map corresponding to the data shown in panel d). The scalebar represents the expected firing rate (Hz) at each point. For visualization purposes the data are copied vertically three times. The red line corresponds to the mean phase as a function of time as explained in panel f).

f) Change in phase for this cell as a function of time. The phase lag value reported at the top of the plot is calculated from raw, unsmoothed data, and is the circular mean phase lag between successive stimuli, in units of radians/stimulus. This phase lag value was computed by taking the circular mean of the (difference between the mean preceding phase and the mean preferred phase) and the (difference between the mean preferred phase and the mean following phase). For display purposes only, the phase value that is plotted is the circular mean phase computed in sliding windows of 2.5s (shifted by steps of 0.112s corresponding to 50 steps in each figure).

In panels d, e, and f), time = 0s is the start of the preferred stimulus, time = -2s is the onset of the preceding stimulus, and time=2s is the onset of the following stimulus. Stimulus onsets are indicated by the thicker vertical lines in panels e) and f), and the ISI onsets are indicated by the thinner vertical lines.

REVIEWER COMMENTS

Reviewer #1 (Remarks to the Author):

The authors have adequately addressed my final points of concern. I feel the manuscript is now ready for publication.

Florian Mormann

Reviewer #2 (Remarks to the Author):

The new details provided by Reddy et al in the revised submission greatly help clarify the analysis and better explain the results, particularly their smoothness, as shown. Unfortunately, these new details reveal issues with the slope determinations that were performed.

In particular, it's clear now that the bins shown in Figure 4, Figure S3f, and Figure S8 are smoothed from between 1.5 to 2.5 s. I appreciate that this point is now explicitly stated in the legends for Figure S3f. I did find this detail also noted around line 500 of the methods as well, but that was rather hard to find. Ideally this should be similarly done in the legends in other figures where it is employed (e.g. Figure 4 and S8). Importantly, this smoothing causes issues for the linear regression that's performed in these plots. Regression is based on the assumption that the mean phase is calculated independently for each timebin, but the smoothing violates that assumption. The 1.5-2.5s duration of the smoothing window also causes additional concerns for bins at the beginning and end of each stimulus representation, since values in those bins are now averaged with values from during other stimuli. Moreover, it's not clear whether the smoothing windows are centered on each timebin, or arranged otherwise. In any case, what happens for bins at the -2s and 3.5 s points? Are these bins smoothed with data for presentation before and after those time points? While the smoothing may be aesthetically appealing, I believe it is creating a misleading impression of the variability in the data and creating statistical problems for the analyses.

For Figure S3, please provide the mean phases calculated for each bin individually (unsmoothed), in either one of the existing panels or a separate panel. Figure 4 incorporates data from a large sample of 452 cells, so this data should not require any smoothing, and each bin should be derived from the mean of the sample at that time point, prior to the linear regression.

I could not understand why data points were missing for some timebins in Figure 4 and Figure S8 (which wasn't the case in the versions of these figures). Please explain.

Currently Figure 3 provides the strongest evidence for phase precession over successive stimuli. This evidence could be strengthened by also providing the mean phase from all the spikes pooled across all neurons (rather than the mean of each cell), in a panel similar to panel B, and applying the Raleigh statistics.

Minor comments:

L170 "continuous" is misleading given the heavy smoothing.

L 230: please provide the relevant degrees of freedom in the statistical test. Presumably around 50 here?

L479 Figure 1C as referenced does not show phase-locking.

Figure S3f. The description of the phase-lag calculation is a bit confusing. Could you please provide a

formula? This makes it seem as if the differential phases are calculated twice for each bin. Would it be simpler to just calculate the mean phase difference between sequential bins?

Please note that there are visible edge effects for the smoothing performed on the Figure S3e panels (e.g. bottom vs. top of cell A, G, etc). But this should not be a problem if unsmoothed data is used in the phase determinations.

Reviewer #2 (Remarks to the Author):

The revised manuscript is much improved. I particularly appreciate that in the supplementary materials for this revision, the authors have now provided some of the raw data for the paper. The new figures, particularly Supp Fig S3, go a long way in helping readers and reviewers understand the quality and features of the data. However, I was surprised by how the spike-theta phase relationships in the sample units were generally weak, particularly in the provided scatterplots. These raise further questions about how they can be reconciled with the provided phase-position slopes in the main figures. Overall, there are some important questions regarding the derivation and reliability of the phase-values central to this study.

--The a panels in Figure S3 examples show filtered and raw traces. The raw traces generally don't appear to have a strong theta component. Apparently the gain is increased on the filtered version (by how much? this should be reported), which makes it difficult to gauge the correspondence. The unfiltered data should ideally be shown at a similar amplification. Also, please provide the phases assigned to the shown spikes. It's hard to see any evidence for phase-precession at the single trial level here.

--Critically, it is difficult to see the correspondence between panels d, e, and f. The scatterplots in d appear generally uniform. There seems to be some heavy smoothing applied to e. How was this smoothing performed? Since Spike Counts are reported, how large are the bins corresponding to these values? The scalebar does not seem to accurately reflect the values apparent by eye (i.e. empty bins show non-zero values, etc.).

--Most concerning is the lack of apparent correspondence with panel f. The authors don't report how the "phase" in these plots was derived. Presumably it is the mean phase? And presumably it was derived from circular statistics? For phase data, the phase should correspond to the angle of the mean resultant vector. I would like to assume that this is how the authors calculated it, but I could not find this specified anywhere. Here again it looks like some kind of smoothing was applied, but details for the smoothing were not provided. In any case the phase-slopes should be provided for raw unsmoothed data. In some examples (e.g. cell C), the smoothed slope in panel f appears to diverge from what can be seen by eye for the density plots in panel e, raising further questions about how the phase values were derived for each time bin. It would be helpful to see how the mean calculated corresponds to the phase histograms from the individual bins, and if f and e were overlaid. It appears that the panel f values were passed into Fig. 4 and Fig S8 analysis, so this is particularly important.

--It's unclear how the mean phase was calculated for Figure 3 as well. It should presumably be the angle corresponding to the mean resultant vector. Obviously, a statistic appropriate to circular data should be used, but I could not find this explicitly stated in the text. I am not sure why smoothing needed to be applied here. Given the large amount of spiking data from a large number of cells, I would assume that the data would already be fairly smooth. The unsmoothed data should also be

provided here (and in related supp figures), in light of the questions already raised.

Reviewer #2 (Remarks to the Author):

The new details provided by Reddy et al in the revised submission greatly help clarify the analysis and better explain the results, particularly their smoothness, as shown. Unfortunately, these new details reveal issues with the slope determinations that were performed.

Point 1:

In particular, it's clear now that the bins shown in Figure 4, Figure S3f, and Figure S8 are smoothed from between 1.5 to 2.5 s. I appreciate that this point is now explicitly stated in the legends for Figure S3f. I did find this detail also noted around line 500 of the methods as well, but that was rather hard to find. Ideally this should be similarly done in the legends in other figures where it is employed (e.g. Figure 4 and S8).

The window size and window number for plotting Figures 4 and S8 are now mentioned in the corresponding figure legends (these figures are included at the end of this letter for your convenience).

We would also like to mention a small, but important, technical distinction here, which is also relevant to some of the other points raised by the reviewer below. In these figures there is no actual smoothing being done. Instead, the mean phase is computed for each bin individually, and the neighboring bins overlap. The overlap does result in non-independent windows, and effectively makes the resulting curves smoother, but there is no actual smoothing operation (e.g. filtering, blurring, sliding average) being performed. However, as the reviewer highlights below, the overlapping bins can still be problematic for linear regression (slope determination), and we have consequently replaced the linear regression test with a Watson-William test performed on non-overlapping data (i.e., similar to the analysis in Figure 3) to compare distributions of phase values across cells. The details are provided below in reply to Point 3.

Finally, we note that our main results are reported in Figure 3, and the main statistical tests were performed on non-overlapping data (lines 150-165). Figures 4 and S8 are supporting figures; however, we do appreciate the reviewer's point and we hope that our replies below alleviate these concerns.

Point 2:

Importantly, this smoothing causes issues for the linear regression that's performed in these plots. Regression is based on the assumption that the mean phase is calculated independently for each timebin, but the smoothing violates that assumption. The 1.5-2.5s duration of the smoothing window also causes additional concerns for bins at the beginning and end of each stimulus representation, since values in those bins are now averaged with values from during other stimuli. Moreover, it's not clear whether the smoothing windows are centered on each timebin, or arranged otherwise. In any case, what happens for bins at the -2s and 3.5 s points? Are these bins smoothed with data for presentation before and after those time points? While the smoothing may be aesthetically appealing, I believe it is creating a misleading impression of the variability in the data and creating statistical problems for the analyses.

Thank you for pointing out that it is not appropriate to perform a linear regression on non-independent data points, and we acknowledge our oversight on this point. In the revised manuscript, we have removed the linear regression and now perform statistical testing on non-overlapping data using the same statistical test (Watson-William test) we used for statistical testing in Figure 3. We describe this analysis in more detail in reply to Point 3 below. We have removed all references to the linear regression. The point about the window size is also discussed in the reply to the point 3.

Finally, we provide the other requested information in the Methods section (page 10):

"In the population-level figures (Figures 4 and S8), for display purposes only, the plotted phase values are the circular mean phase values across cells computed in sliding windows (steps of 0.112s corresponding to 50 steps in each figure). The sliding windows were centered on each

time bin, and any window that expanded beyond the trial edges (i.e., -2s and 3.5s) were truncated accordingly.”

Point 3:

For Figure S3, please provide the mean phases calculated for each bin individually (unsmoothed), in either one of the existing panels or a separate panel. Figure 4 incorporates data from a large sample of 452 cells, so this data should not require any smoothing, and each bin should be derived from the mean of the sample at that time point, prior to the linear regression.

As we mentioned above, there is no actual smoothing operation being performed. The mean phases are computed in individual bins, and the bins overlap, effectively smoothing the curves. In Figures 4, S3, and S8 we now provide the mean phase for windows that are non-overlapping.

About the bin or window resolution at which the mean phase is computed in Figure 4 (and Figures S3f and S8), the mean phase value cannot be computed as an instantaneous value, but requires integration over a time bin or window. The question then boils down to determining an acceptable window size -- because of the low firing rate of our cells, in small bins the mean phase might be undefined since there may be no spikes in the bin. It thus becomes necessary to make a compromise between window size and firing rates, while also considering the number of cells in each analysis. We discuss this point in more detail below.

Determination of an appropriate window size:

For the population level analyses, we are constrained to using relatively large time windows because we do not pool spikes across cells (otherwise the high firing rate cells would dominate). Instead, each cell contributes one point in each time bin. Because the firing rate of (some of) our cells is quite low, the phase estimate in small time bins would be unreliable. For instance, during a 100ms bin, 87% of cells would fire less than 15 spikes so that a proper estimate of mean phase could not be computed. Conversely, with bin sizes of 1s or 1.5s, 70-80% of cells fire more than 15 spikes in the window.

In the previous versions of our paper, the window size for Figure 4 was 1.5s because we pooled over a large number of cells (452 cells; as mentioned in the Methods section). For Figure S8 we used a median split (~225 cells) and the window size 2s. For the remaining analyses (e.g., Figure S3f for individual cells, or S8A where the split includes only ~100 cells) the window size was 2.5s. The reviewer's comment prompted us to launch a proper statistical power analysis, which confirmed our observations.

We performed a new simulation to determine the appropriate window size for our data as a function of the number of cells and their firing rates. In the simulation, we considered the true firing rate of every cell, the preferred phase, and concentration parameter (kappa value). We then generated Nspk random angles from a von Mises distribution given that cell's preferred phase and concentration parameter (we used the `circ_vmrnd` function in the `CircStats` Toolbox). The value of Nspk was determined from the firing rate of each cell. We simulated the population to determine the reliability of phase estimates as a function of window size. Specifically, for a given criterion (e.g. $\pm \pi/8$, which would allow us to reliably detect a phase difference of 45 degrees), we asked how likely it is that the simulated mean phase across cells in the population lies within \pm criterion of the real mean phase of the population, for each window size.

The population of 452 cells (as in Figure 4) was simulated 5000 times, each time with a new random distribution of spikes for each cell, for window sizes of 0.5s, 1s, 1.5s, 2s, 2.5s, 3s and 4s. In each iteration, we compared the population simulated mean phase and the real mean phase of the population. The probability of success (i.e., that the simulated and real mean phase values lie within the criterion) as a function of window size is shown in Figure R1 below.

These simulation results show that with a population of 452 cells, it takes a window size of 1.5s for the population mean phase to be reliably determined 80% of the time (i.e., to reach 80% statistical power).

Figure R1: Simulation results for a population of 452 cells for different window sizes. The probability of success on the y-axis corresponds to the probability that the simulated phase of the population and the real mean phase of the population lie within a distance of \pm criterion.

We applied the same logic to determine the window size for the analyses in Figure S8 where the population size was reduced by half (because of the median split). Because the number of cells was smaller in this control analysis, we achieved a power of 0.7 with a window size of 2s.

Figure R2: Simulation results for random halves of our population for different window sizes. The probability of success on the y-axis corresponds to the probability that the simulated phase of the population and the real mean phase of the population lie within a distance of \pm criterion.

We repeated this analysis at the single cell level. For each cell we performed 1000 simulations taking into account its real firing rate and a given window size to determine the number of spikes in each window (NSpk). We then drew NSpk spikes based on the median kappa of the population, and from the NSpk spikes we estimated the cell's preferred phase. The phase estimate was considered accurate if it lay within \pm criterion_{simulation} of the true mean phase (we set criterion_{simulation} = $\pi/4$ meaning that the phase estimate is considered accurate when it falls within the same quadrant as the true phase).

If at least half of the simulations produced an accurate phase estimate, we considered the cell successful. The graph below (Figure R3) shows the proportion of successful cells as a function of window size. Based on this simulation, we selected the maximum non-overlapping window size of 2.5s for phase computations.

Figure R3: Simulation results at the single cell level. The figure shows the proportion of successful cells as a function of window size.

These simulations justify the window sizes that we had empirically chosen.

Replacing the linear regression analysis with the Watson-William test:

As the reviewer noted, computing a linear regression slope with overlapping bins is problematic. We have thus replaced all linear regression tests with a pairwise Watson-William test for equal means performed on non-overlapping data. This is the same test that we have used in our main results (Figure 3), and we believe that the use of this test for the supporting analyses makes the paper more coherent. Specifically, in Figures 4 and S8, we perform the Watson-William test on the phase distributions in non-overlapping bins for the extreme stimuli (i.e., the preceding vs. the following stimulus). The figure panels show the p-values of the Watson-William test computed over these two bins (the other test details are in the figure legend). To provide an estimate of the strength of phase precession over the duration of the trial we have replaced the regression slope measure with the actual phase difference or lag between these two bins in units of degrees/stim:

$$Phase\ Lag = (mean_{prec} - mean_{fol})/2$$

This change is mentioned in each figure legend (copied at the end of this letter for your convenience), and also in the Methods section (Page 10):

“Phase precession lags were computed by taking the difference of the mean phase across cells in non-overlapping windows for the extreme stimuli (i.e., the preceding vs. the following stimulus), and statistical significance was determined with a Watson-William test.”

Providing mean phases in individual, non-overlapping bins:

Finally, the reviewer requested that we show the mean phase values in individual, non-overlapping bins. Based on the simulation results, this would correspond to showing phase values in two or three non-overlapping time bins in each panel (the number of non-overlapping time bins depends on the window size, W). We now indicate these mean phase values as large white dots in each panel (note that the Watson-William test is computed on these non-overlapping values).

These changes are shown at the end of this letter for Figures 4, S3 and S8.

Point 4:

I could not understand why data points were missing for some timebins in Figure 4 and Figure S8 (which wasn't the case in the versions of these figures). Please explain.

We only plot data in time bins for which the requirements for the confidence limits are met, as determined by the Circular Statistics toolbox (these requirements are described in Ref 49). In the previous version of these figures, the mean values for these points had been plotted, but without error bars. However, this plotting detail was potentially misleading about the variability in our data, and for the sake of clarity, we have now chosen to refrain from plotting these points. This is now mentioned in the figure legends.

Point 5:

Currently Figure 3 provides the strongest evidence for phase precession over successive stimuli. This evidence could be strengthened by also providing the mean phase from all the spikes pooled across all neurons (rather than the mean of each cell), in a panel similar to panel B, and applying the Raleigh statistics.

Thank you for this suggestion. We provide this data in Figure S2 (F). These results are copied below for your convenience. This figure is referenced to in the figure legend for Figure 3.

Figure R4: Circular histogram when pooling all spikes from the population ($N=449$, i.e., 2 cells with firing rate greater than 5 standard deviations above the mean firing rate of the population were removed since outlier cells with exceptionally high firing rates could dominate the outcome). The dashed colored lines indicate the mean phase, and the colored angular lines correspond to one standard deviation. The p-values above the rose plots are the results of a two-sample Watson-William test for equal means. Results of the Watson-William test: ($F(1,99475) = 198.8$; $p < 0.0001$ for preferred vs. preceding; $F(1,99605) = 404.9$; $p < 0.0001$ for preferred vs. following; and $F(1,90939) = 990.7$; $p < 0.0001$ for preceding vs. following).

Minor comments:

Point 6: L170 “continuous” is misleading given the heavy smoothing.

We have removed the word “continuous”. The description for Figure 4 in the text now reads as follows (Page 4):

“We observed a change in the phase during successive stimuli. The phase difference between the two extreme stimuli (preceding vs. following) was significant in the theta band (60%stimulus, Watson-William test, $F(1,903) = 48.3$, $p < 0.001$), and just significant in the beta band (12%stimulus, Watson-William test, $F(1,903) = 5.5$, $p < 0.05$).”

Point 7: L 230: please provide the relevant degrees of freedom in the statistical test. Presumably around 50 here?

This linear regression analysis has been replaced as explained in Point 3. The results of the Watson-William test are now provided.

Figure 4 legend:

“The phase lag was significant in the theta band (60 %stimulus, Watson-William test, $F(1,903) = 48.3, p < 0.001$), and just significant in the beta band (12 %stimulus, Watson-William test, $F(1,903) = 5.5, p < 0.05$).”

Point 8: L479 Figure 1C as referenced does not show phase-locking.

Thank you for pointing this out. We now refer to Figure 2C.

Point 9: Figure S3f. The description of the phase-lag calculation is a bit confusing. Could you please provide a formula? This makes it seem as if the differential phases are calculated twice for each bin. Would it be simpler to just calculate the mean phase difference between sequential bins?

To be consistent throughout the paper we now apply the same measure for phase lag in Figures 4, S8 and S3.

Figure S3f legend:

“The phase lag value (in units of radians/stim) reported at the top of the plot is calculated from these non-overlapping bins as follows: $Phase\ Lag = (mean_{prec} - mean_{follow})/2$.”

Point 10: Please note that there are visible edge effects for the smoothing performed on the Figure S3e panels (e.g. bottom vs. top of cell A, G, etc). But this should not be a problem if unsmoothed data is used in the phase determinations.

Thank you for pointing this out. Yes, the phase lag values are computed on unsmoothed data.

Figure 4: Phase as a function of time in the theta (left) and beta (right) bands. The blue, pink and yellow shaded areas correspond to the stimulus periods of the preceding, preferred and following stimuli. The white regions between stimuli are inter-stimulus intervals. For visualization purposes only, the black dots and lines correspond to the circular mean phase and standard deviation across cells computed in sliding windows of 1.5s. The circular mean was only plotted when the requirements for confidence limits were met. The large white points correspond to the circular mean phase in (non-overlapping) bins centered on the preceding, preferred, and following stimuli. The phase lag value (in units of degrees/stim) reported at the top of each panel is calculated from non-overlapping bins as follows: $Phase\ Lag = (mean_{prec} - mean_{fol})/2$. Significance of the phase lag was determined by a Watson-William test on the phase distributions for the preceding vs. following stimuli. The gray dashed lines correspond to the best linear fit through the data, and is plotted for illustration purposes. The phase lag was significant in the theta band ($60^\circ/\text{stimulus}$, Watson-William test, $F(1,903) = 48.3$, $p < 0.0001$), and weakly significant in the beta band ($12^\circ/\text{stimulus}$, Watson-William test, $F(1,903) = 5.5$, $p < 0.05$).

Following pages: Figure S3. (A-L). Examples of individual cells showing phase-precession. The format is the same for all cell examples.

a) 3-seconds of raw and filtered (4-8Hz) LFP traces in a single example trial. The tick marks correspond to simultaneously recorded spikes.

b) Raster plots and instantaneous firing rates for the preceding (blue), preferred (pink), and following (yellow) stimuli. The lighter areas correspond to the 0.5s inter-stimulus interval (ISI). The darker areas correspond to the 1.5s image presentation window. Image onset occurred at time=0s. For each cell, the preferred stimulus was the one that elicited the largest number of spikes across all repetitions of the sequence, and the consistency of stimulus preference was verified with a cross-validation analysis. The consistency score from the cross-validation analysis is reported in the yellow panel (see the methods section for a description of the cross-validation analysis and the consistency score). For all cells shown here, the consistency score was significantly higher than chance (0.5, Binomial test, $p < 0.001$). The black curves and grey shaded areas correspond to the mean and standard error of the mean of firing activity across trials. For determining stimulus preferences all repetitions of each stimulus were included in the analysis, excluding the probe trials. Since the probe events occurred randomly, the number of non-probe trials for each stimulus can be different, resulting in different numbers of trials in the raster plots.

c) Phase distribution of spikes in the 4-8Hz theta band for this cell. The significance was determined by a Rayleigh test.

d) Scatter plot of theta phase versus time. For visualization purposes the phase values are copied vertically three times. The red line corresponds to the mean phase as a function of time as explained in panel f).

e) Heat map corresponding to the data shown in panel d). The scalebar represents the expected firing rate (Hz) at each point. For visualization purposes the data are copied vertically three times. The red line corresponds to the mean phase as a function of time as explained in panel f).

f) Change in phase for this cell as a function of time. For display purposes only, the phase value that is plotted in the black curve is the circular mean phase computed in sliding windows of 2.5s (shifted by steps of 0.112s corresponding to 50 steps in each figure). The larger black points and error bars correspond to the mean and standard deviation of the phase in (non-overlapping) bins for the preceding and following stimuli. The phase lag value (in units of radians/stim) reported at the top of the plot is calculated from these non-overlapping bins as follows: $Phase\ Lag = (mean_{prec} - mean_{fol})/2$. In panels d, e, and f), time = 0s is the start of the preferred stimulus, time = -2s is the onset of the preceding stimulus, and time=2s is the onset of the following stimulus. Stimulus onsets are indicated by the thicker vertical lines in panels e) and f), and the ISI onsets are indicated by the thinner vertical lines.

Figure S3 continued.

Figure S3 continued.

Figure S3 continued.

Figure S3 continued.

Figure S3 continued.

K.

L.

Figure S3 continued.

Figure S8. Strength of phase precession examined in different conditions. In all panels, for display purposes only, the small black dots and lines correspond to the circular mean and standard deviation of phase values across cells, plotted as a function of time in sliding windows (Methods). The circular mean was only plotted when the requirements for confidence limits were met. The mean phase was computed in sliding time windows; as described in the Methods section in panels where the data were reduced by a median split (B, E-G), the window size was 2s, and 2.5s for the remaining panels. The large white points and error bars correspond to the mean and standard deviation of the phase in (non-overlapping) bins for the preceding and following stimuli. The phase lag value (in units of degrees/stim) reported at the top of each panel is calculated from these non-overlapping bins as follows: $Phase\ Lag = (mean_{prec} - mean_{fou})/2$. Significance of the phase lag was determined by a Watson-William test on the phase distributions in these non-overlapping windows. The gray dashed lines correspond to the best linear fit through the data, and is plotted for illustration purposes when the phase lag is significant. A) The neurons were divided into a group of cells that were “strongly-tuned” (see Methods) as determined via a one-way ANOVA of the firing response across the stimuli presented in the sequence ($p < 0.05$). The remaining “weakly-tuned” cells were cells with a weaker, but reliable response as determined by a non-parametric cross-validation analysis. In both groups, the phase lag was significantly negative (left, $F(1,203) = 12.5, p < 0.001$; right $F(1,699) = 52.6, p < 0.0001$). B) The neurons were split into two groups according to the median phase locking strength. The phase-locking for each cell was

determined outside the [preceding, preferred, following] window, since phase precession effects in this window might be expected to negatively affect phase-locking. Thus, the strength of phase-locking was determined based on spikes fired to the preceding-1 and following+1 stimuli (the phase values for these stimuli were similar, Figure S2). In both the low and high phase-locking groups, the phase lag was significantly negative (left, $F(1,449) = 17.5, p < 0.0001$; right $F(1,453) = 33.5, p < 0.0001$). C) Cells were split according to the length of the stimulus sequence (left, $F(1,389) = 5.7, p < 0.05$; right $F(1,485) = 52.6, p < 0.0001$). D) Cells were split based on the location of the electrodes (left, $F(1,723) = 78.4, p < 0.0001$; right $F(1,179) < 0.01, p = 0.9$). E) Cells were split into two groups according to the median F-value from a one-way ANOVA of the firing response across all stimuli presented in the sequence (left, $F(1,459) = 9.6, p < 0.005$; right $F(1,443) = 11.1, p < 0.001$). F) The data for each cell was split according to the first half or second half of trials (early learning vs. later learning; left, $F(1,903) = 68.9, p < 0.0001$; right $F(1,903) = 120.3, p < 0.0001$). G) Stimulus selectivity for each cell was determined by taking the ratio of the mean response to the “preferred” stimulus vs. the mean response to the “other” stimuli. Cells with strong selectivity for their preferred stimulus, such as concept cells (Quiroga, 2012) have a low mean response to the other stimuli and therefore higher selectivity. Cells were split into two groups according to the median value of this measure of stimulus selectivity (left, $F(1,461) = 38.6, p < 0.0001$; right $F(1,441) = 18.2, p < 0.0001$). The phase lag tended to be weaker for the more selective cells, but the difference was not significant between the two groups.

REVIEWER COMMENTS

Reviewer #2 (Remarks to the Author):

I appreciate the authors' response to my comments, which have now all been addressed. I believe in its current form the manuscript is both interesting and informative.

A couple of minor comments: I am a bit confused by the x-axis alignment of the mean phases in Fig S3 and S8. From the variations in S8 I would assume that these correspond to the mean time-points of the spikes, but that doesn't seem to be correct for S3.

Please note that the x-tick labels on Figure S8 are misaligned in panel F.

Reviewer #2 (Remarks to the Author):

I appreciate the authors' response to my comments, which have now all been addressed. I believe in its current form the manuscript is both interesting and informative.

Thank you for your time in providing comments on the manuscript. We are pleased that we have been able to address all your comments.

A couple of minor comments: I am a bit confused by the x-axis alignment of the mean phases in Fig S3 and S8. From the variations in S8 I would assume that these correspond to the mean time-points of the spikes, but that doesn't seem to be correct for S3.

The mean phase is plotted at the center of the time window in which it was computed. For example, in Figure S3, the window size was 2.5s, so the mean phase for the first (preceding) stimulus is plotted at the center of the [-2 0.5] s window, i.e., at -0.75s. We specify this in the figure legends.

Figure S3f: The larger black points and error bars correspond to the mean and standard deviation of the phase in (non-overlapping) bins for the preceding and following stimuli, plotted at the center of the 2.5s window in which the mean phase was computed.

Figure S8: The large white points and error bars correspond to the mean and standard deviation of the phase in (non-overlapping) bins for the preceding and following stimuli, plotted at the center of the time window in which the mean phase was computed.

Please note that the x-tick labels on Figure S8 are misaligned in panel F.

Thank you for pointing this out. We have corrected the x-tick labels.